# Partial FAM19A5 deficiency in mice leads to disrupted spine maturation, hyperactivity, and an altered fear response

Anu Shahapal[1], Sumi Park[1], Sangjin Yoo[2], Shi-Xun Ma[2], Jongha Lee[2], Hoyun Kwak[2], Jong-Ik Hwang[1], Jae Young Seong[1,2]*

1 Department of Biomedical Sciences, Graduate School of Medicine, Korea University, Seoul, Republic of Korea, 2 Neuracle Science Co., Ltd., Seoul, Republic of Korea

* jyseong@korea.ac.kr

## Abstract

The FAM19A5 polypeptide, encoded by the TAFA5 gene, is evolutionarily conserved among vertebral species. This protein is predominantly expressed in the brain, highlighting its crucial role in the central nervous system. Here, we investigated the potential roles of FAM19A5 in brain development and behavior using a FAM19A5-LacZ KI mouse model. This model exhibited a partial reduction in the FAM19A5 protein level. FAM19A5-LacZ KI mice displayed no significant alterations in gross brain structure but alterations in dendritic spine distribution, with a bias toward immature forms. These mice also had lower body weights. Behavioral tests revealed that compared with their wild-type littermates, FAM19A5-LacZ KI male mice displayed hyperactivity and a delayed innate fear response. These findings suggest that FAM19A5 plays a role in regulating spine maturation and maintenance, thereby contributing to neural connectivity and behavior.

## Introduction

FAM19A5, a polypeptide encoded by the TAFA5 gene located on chromosome 22 (22q13.32) in humans and chromosome 15 in mice, is expressed primarily in the brain [1]. This restricted expression and the highly conserved amino acid sequence across vertebrates suggest a crucial role for FAM19A5 in brain development and function [2,3]. A study by Huang et al. showed that deletion of FAM19A5 led to reduced spine density, glutamate signaling and neuronal activity. These changes resulted in depressive-like behavior and impaired spatial memory in mice, suggesting that FAM19A5 acts as a key regulator of early brain development and cognitive function [4].

Additionally, a few human cases of developmental defects associated with deletions in chromosome 22, where FAM19A5 is located, have been reported [5,6]. The microdeletion of chromosome 22, del22q13.32-q13.33, encompasses exon 4 of

**Data availability statement:** The underlying data for the main and supplementary figures are available from Figshare at the following DOIs: https://doi.org/10.6084/m9.figshare.29289002 and https://doi.org/10.6084/m9.figshare.29289056, respectively. Full datasets supporting the findings of this study will be available upon reasonable request from the corresponding author.

**Funding:** This research was supported by internal company funding from Neuracle Science Co., Ltd. and by the Research Program of the National Research Foundation of Korea (NRF-2020M3E5D9080794), which is funded by the Korea Government (MSIT). The funders had no role in study design, data collection and analysis, decision to publish, or preparation of the manuscript.

**Competing interests:** S.A, SM.P, and JY.S are shareholders of Neuracle Science Co., Ltd. SJ.Y, HY.K and JH.L are employees of Neuracle Science. There are no patents, products in development or marketed products associated with this research to declare. This does not alter our adherence to PLOS ONE policies on sharing data and materials.

FAM19A5 and is accompanied by clinical features such as hyperactivity, aggression, low body weight, skeletal abnormalities, speech disorders, and brain deformities. This finding suggested that loss of FAM19A5 function disrupts critical aspects of brain development and function, potentially contributing to a range of behavioral and cognitive impairments. However, the specific mechanisms by which FAM19A5 deletions lead to these developmental issues remain poorly understood, necessitating further investigation into its role in neurodevelopment.

Previously, we investigated the potential function of FAM19A5 via a FAM19A5-LacZ KI mouse model [3,7]. FAM19A5-LacZ knock-in (KI) mice displayed LacZ expression patterns consistent with the FAM19A5 mRNA levels observed via in situ hybridization of both embryonic and adult brains. Additionally, this LacZ expression aligns with recent single-cell RNA sequencing data from mouse and human brains [3,7–9], suggesting successful integration of the LacZ gene. Recently, we observed that crossbreeding FAM19A5-LacZ KI mice with APP/PS1 mice, a mouse model of Alzheimer's disease (AD), extended the lifespan of APP/PS1 mice. Furthermore, systemic administration of monoclonal antibodies against FAM19A5 significantly improved cognitive behavior in APP/PS1 mice [10,11], indicating FAM19A5 as a potential therapeutic target. Molecular studies have shown that FAM19A5 binds with high affinity to a domain within LRRC4B, a post-synaptic adhesion molecule. This interaction likely modulates the synaptic structure, leading to synapse elimination [11]. Thus, FAM19A5 may function as a synaptolytic factor under physiological conditions, while also contributing to excessive synapse loss in neurological diseases such as AD. In this context, inhibiting FAM19A5 function has been demonstrated to restore mature spine density, resulting in functional recovery of synapses and cognition in mouse models of AD [10,11].

Here, we further investigated the potential role of FAM19A5 in brain development using a FAM19A5-LacZ KI mouse model [3,7]. FAM19A5-LacZ KI mice exhibited reduced levels of the FAM19A5 protein in the brain, and the mutant FAM19A5 protein generated through this gene modification showed a lower binding affinity for its binding partner, LRRC4B [7,11], suggesting that this model functions as a partial knock-out (KO) of FAM19A5. We examined physical deformities, dendritic spine morphology and behavioral changes in FAM19A5-LacZ KI mice, providing insights into the role of FAM19A5 in the regulation of brain functions.

## Materials and methods

### Animals

All the mice were housed under temperature-controlled (22–23°C) conditions with a 12-h light/12-h dark cycle (lights on at 8:00 am). The mice were given ad libitum access to standard chow and water. All the animal experiments were designed to use the fewest mice possible, and anesthesia was administered before sacrificing the mice. Anesthesia was induced by placing the mice in a clean induction chamber and delivering 3% isoflurane. All animal procedures were approved by the Institutional Animal Care and Use Committee of Korea University (KOREA-2016–0091-C3).

Homozygous FAM19A5-LacZ KI mice were obtained by mating heterozygous male and female FAM19A5-LacZ KI mice. WT littermates were used as the control group. Body weights were measured weekly starting from four to ten weeks after birth. The brain sizes were measured during brain sampling using a digital Vernier caliper. The daily food intake of six-week-old male mice was measured for 15 days. For the behavior test, ten twelve-week-old male mice were used unless otherwise mentioned.

## Nissl staining and measurement of cortical thickness/layer

Three-month-old mice were perfused with 4% paraformaldehyde in PBS (pH 7.4), and the isolated brains were post-fixed in the same solution overnight at 4°C. The brains were then cryo-protected in 30% sucrose in PBS, serially cross-sectioned into 40-μm sections using Cryostat (Leica), and stored in 50% glycerol in PBS at −20°C. The coronal brain sections were dipped in 1% cresyl violet solution (C5042, Sigma-Aldrich, Missouri, United States) at 50°C for 5–10 min. The sections were rinsed with distilled water, dehydrated, and mounted with Permount (SP15–500, Thermo Fisher Scientific, Massachusetts, United States). Nissl-stained sections were imaged using a slide scanner (AxioScan Z1, Zeiss). Different brain regions were defined using the mouse brain atlas (motor cortex, bregma 1.34~0.98 mm; somatosensory cortex, bregma −1.22~−1.58 mm; visual cortex, bregma −2.54~2.92 mm; and auditory cortex, bregma −2.54~2.92 mm) [12]. For cortical layer measurements, different layers were distinguished based on the cellular morphology and arrangement [12,13]. Three nonadjacent Nissl-stained brain sections at intervals of 120 μm from each genotype (n = 5–6) were analyzed. The average cortical thickness and layer thickness were calculated from three independent linear measurements using ZEN software (Zeiss).

## Golgi staining and dendritic spine analysis

Ten-week-old mouse brains from equal numbers of male and female mice were subjected to spine analysis. Golgi staining was performed via the FD Rapid GolgiStain Kit (FD NeuroTechnologies) following the manufacturer's protocol. Briefly, mouse brains were isolated and immersed in impregnation solution (a mixture of solutions A and B) for two weeks in the dark at room temperature. Next, the brains were transferred to Solution C and stored at room temperature in the dark. Coronal sections 100 μm thick were obtained via a vibratome and transferred to gelatin-coated slides (Lab Scientific). The sectioning procedure was carried out in Solution C to prevent cracking of the sections. Solution C was blotted completely from the slide, and the sections were then air-dried at room temperature overnight, washed with DW three times every 5 min and subjected to working solution (a ratio of 1:1:2 of solution D:E:DW). After being washed in DW 3 times every 5 min, the sections were dehydrated in increasing concentrations of ethanol (50%, 75%, 90% and 100%) for 5 min each, cleared in xylene and coverslipped with Permount solution. Z-stack images (0.5 μm intervals) of the basal and apical dendrites of approximately 5–7 pyramidal neurons in layers 2/3 and layer 5 of the motor cortex and somatosensory cortex were acquired using a 60X objective with a confocal microscope (TCS SP8, Leica). Spines from secondary basal and apical dendrite segments of at least 20 μm in length were selected for quantitative analysis. Approximately 5 dendritic segments from each neuron were analyzed. The classification of spines and the width and length of an individual spine were manually measured as previously described [14]. The spine density was expressed as the number of spines/10 μm of dendrite. The length-to-width ratio (LWR) was calculated as a measure of spine morphology.

## Quantitative real-time polymerase chain reaction

TRI Reagent was used to extract total RNA from mouse brain tissue. cDNA was synthesized from 1 μg of RNA using the RevertAid First Strand cDNA Synthesis Kit. The following primer sequences were used for quantitative RT-PCR: mFAM 19A5-F, 5′-AGG TGA ATG ACC CCC TTC GT-3′; mFAM19A5-R, 5′-TGA CTC TGC TCC CCA GCT TC-3′; mGAPDH-F, 5′-AAG GTC ATC CCA GAG CTG AA-3′; mGAPDH-R, 5′-CTG CTT CAC CAC CTT CTT GA-3′. Quantitative RT-PCR was

conducted on a CFX96 Touch™ Real-Time PCR Detection System with SsoAdvanced Universal SYBR® Green Supermix. FAM19A5 and GAPDH expression levels were measured using specific primers. Gene expression was normalized to that of GAPDH and analyzed via the comparative Ct method.

## Cell culture and transfection

HEK293 cells were cultured in Dulbecco's Modified Eagle Medium (DMEM) supplemented with 10% fetal bovine serum (FBS) and 1% penicillin-streptomycin, and maintained at 37°C in a humidified incubator with 5% $CO_2$. For the overexpression of wild-type (WT) and mutant (MT) FAM19A5, the cells were transfected using the Neon Transfection System (Invitrogen) according to the manufacturer's instructions.

## Protein stability assay

Transfected HEK293 cells were treated with 100 μg/mL cycloheximide (Sigma-Aldrich) and 10 μM MG132 (Sigma-Aldrich) to inhibit protein synthesis and proteasome activity, respectively. The cells were harvested at 0, 3, 6, and 9 hours post-treatment for further analysis. At each time point, the cells were washed twice with ice-cold phosphate-buffered saline (PBS) and lysed in NP-40 lysis buffer (50 mM Tris-HCl, pH 7.5, 150 mM NaCl, 1% NP-40) supplemented with protease and phosphatase inhibitors on ice for 30 minutes. The lysates were then cleared by centrifugation at 15,000 rpm for 30 minutes at 4°C.

The Western blot membranes were acquired as 16-bit grayscale TIFF images using a ChemiDoc MP Imaging System (Bio-Rad), with identical exposure settings applied to every lane. Band intensities were quantified in ImageJ (Fiji distribution) by measuring the integrated density of each target protein band and the corresponding loading control. For each lane, the target protein's net intensity (background-subtracted) was divided by the net intensity of its loading control. The resulting normalized value at time 0 was set to 1.0, and all subsequent time points were expressed relative to this baseline to generate time-dependent protein decay curves.

## Co-immunoprecipitation

HEK293 cells were transfected with the FAM19A5 and FLAG-LRRC4B plasmids using the Neon™ Transfection Kit (Invitrogen) following the manufacturer's instructions. Post-transfection, the cells were washed twice with cold phosphate-buffered saline (PBS) and lysed in buffer containing 20 mM Tris-HCl (pH 7.4), 150 mM NaCl, 0.5% NP40, and a protease and phosphatase inhibitor cocktail (Thermo Scientific). The lysates were subsequently centrifuged at 12,000 × g at 4°C for 30 minutes to obtain the supernatant. Protein G Dynabeads (Invitrogen), pre-incubated with antibodies, were added to the supernatant, which was subsequently incubated overnight at 4°C on a rotating mixer. The beads were then washed three times with washing buffer containing 20 mM Tris-HCl (pH 7.4), 300 mM NaCl, and 0.5% NP40. Finally, the bound proteins were eluted by boiling the beads in 2 × sample buffer containing a reducing agent for 10 minutes and subsequently separated by SDS-PAGE.

## ELISA

To measure FAM19A5 in CSF, we utilized the established enzyme-linked immunosorbent assay (ELISA) used in a previous study [10]. Briefly, the 96-well microplates were coated with LRRC4B (453–576) protein and diluted in 50 mM carbonate buffer (pH 9.6) to a final concentration of 1 μg/ml. After overnight incubation at 4°C, the plates were washed twice with washing buffer (PBS with 0.05% Tween 20) and blocked with blocking buffer (PBS with 1% BSA and 0.05% Tween 20) for 1 hour at 37°C. Standard solutions and samples were added to the wells, followed by incubation and additional washes. HRP-conjugated C-A5-Ab (0.2 μg/ml) was added, and the mixture was incubated at 37°C. TMB solution (Thermo Scientific) was then added, followed by the addition of sulfuric acid to stop the reaction. The optical density (OD) was measured at 450 nm via a microplate reader (Molecular Devices).

## Western blot analysis

The mice were decapitated, and their brains were isolated immediately and chilled in ice-cold PBS. The brain tissues were homogenized in lysis buffer (N-PER™ Neuronal Protein Extraction Reagent, Thermo Scientific) containing a protease inhibitor cocktail. The homogenate was centrifuged for 10 min at 15,000 rpm, after which the supernatant was obtained. The protein concentration of the supernatant was determined using a Bradford protein assay kit (Bio-Rad). The brain lysates were then denatured in SDS sample buffer and separated by SDS–PAGE. The proteins were transferred to a polyvinylidene difluoride (PVDF) membrane and blocked with a 5% skim milk solution for 30 min at room temperature. The membrane was incubated with a FAM19A5 3–2-HRP-conjugated antibody (Neuralcle Sciences, 20 μg) overnight at 4°C. After washing with TBST 3 times every 10 min, the signals were detected using an enhanced chemiluminescence (ECL) assay kit (Thermo Scientific). β-Actin (Abcam) was used to normalize the levels of protein detected.

## Open field test

The open field test (OFT) was used to assess exploratory behavior and general locomotor activity. The mice were placed in a white opaque open field box (40×40×40 cm) and allowed to explore freely for 10 min. The field was divided into two zones, the central and peripheral zones. The zone 10 cm from the edge of the field was defined as the center zone. The total distance traveled, mean speed of movement and total time spent in the center of the field were analyzed via ANY-Maze 6.36 software (Stoelting). In addition, ethological parameters such as rearing and grooming were analyzed manually.

## Elevated plus maze test

The elevated plus maze test (EPMT) was used to measure anxiety-like behavior. The maze consisted of two oppositely positioned open arms (5×30 cm) and an oppositely positioned closed arm (5×30 cm) enclosed by a 20 cm high wall placed 50 cm above the ground. The mice were placed in the center of the maze facing the open arm and allowed to freely explore the maze for 10 min. The total time spent in the open arms, total distance traveled and mean speed of movement in the maze were analyzed using ANY-Maze 6.36 software (Stoelting). Arm entry was considered only when all four paws were inside the defined zone.

## Novel object recognition test

The novel object recognition (NOR) test was used to assess recognition memory. The test was conducted in an open field box (40×40×40 cm) with two distinct objects (Lego block and T-75 flasks filled with sand). The mice were habituated to the open field without any objects for 10 min one day before the training session. On the next day, during the training phase, the mice were allowed to freely explore the field containing two identical objects for 10 min. The test phase was performed after 6 h for short-term memory and after 24 h for long-term memory. In the test session, one of the objects was replaced with a novel object, and the mice were placed in the field for 10 min. The time spent exploring each object was analyzed using ANY-Maze 6.36 software (Stoelting). Preference for the novel object was defined as the time spent exploring the novel object divided by the total time spent exploring both objects.

## Y-maze test

The Y-maze test was used to measure spatial learning memory. The mice were placed in the center of the Y-shaped maze with three identical arms (30×5×20 cm), and movement was recorded for 5 min. The total number of arm entries and sequence of entries were analyzed using 6.36 software (Stoelting). Entry occurs when all four limbs are within the arm. Spontaneous alteration was defined as the number of triads (ABC, ACB, BAC, BCA, CAB, CBA) divided by the total number of arm entries minus 2 multiplied by 100.

## Marble burying test

The marble burying test was used to study repetitive behavior and anxiety-like behavior. A cage containing a total of 20 marbles arranged on the surface of 5 cm thick bedding in 5 rows of 4 marbles was employed. The mice were placed in cages containing marbles and left undisturbed for 30 minutes. The total number of marbles buried was counted and plotted as a graph. Marbles were considered buried if 2/3rd or more of the surface area was covered by bedding. One experimenter scored the number of marbles buried.

## Nestlet shredding test

The nestlet shredding test was used as a measure of repetitive compulsive-like behavior. Each nestlet was weighed and placed in a cage with 0.5 cm thick bedding. The mice were placed into the cage for 30 min without any disturbance. Food and water were also provided during the test period. After the test was complete, the nestlets were allowed to dry overnight to avoid moisture. The next day, the unshreded nestlets were weighed, and the percentage of shredded nestlets was plotted.

## Rotarod test

The rotarod test was performed to assess motor learning and coordination. The mice were placed on a rotating rod at increasing speeds (from 4 to 40 rpm) for 5 min. The test was carried out for 5 consecutive days, and the latency to fall and the time it took the mice to fall off the rotating rod were recorded. The mice were subjected to three trials per day, with a maximum of 300 s for each trial and an interval of at least 15 min between trials.

## Hanging wire test

The hanging wire test was used to measure muscle strength, mainly on the forelimbs and hindlimbs. The mice were placed on top of the wire mesh, which was then inverted, and the mice were allowed to hang for a maximum of 300 s. Five repetitive trials were performed in which the first 2 trials involved acclimatization, and the latency to fall during the remaining 3 trials was recorded, the highest of which were plotted.

## Tail suspension test

The tail suspension test (TST) was used to measure depressive-like behavior. The mice were suspended above the ground by the tail with the aid of adhesive tape for 6 min. The movement of the mice was monitored, and immobility was measured using ANY-Maze 6.36 software (Stoelting). Immobility was defined as the time at which the mice stayed without body movement.

## Pavlovian fear conditioning test

The mice were habituated to the fear conditioning chamber for 10 min without any stimulus. The next day, a training session was performed in which 7 repetitive trials, each consisting of a tone (5 kHz, 70 dB, 30 s) that terminated with a foot shock (0.7 mA, 2 s), were delivered. The mice were returned to their home cages, and the test session was conducted the following day. For the contextual memory test, the mice were placed in the same chamber for 5 min without tone or electric shock. The freezing time was analyzed using ANY-Maze 6.36 software (Stoelting). For the auditory memory test, the mice were placed in a distinct context and exposed to 3 tones without foot shocks with an intertrial interval of 90 s after 5 min of acclimatization. The freezing time during each tone was analyzed using ANY-Maze 6.36 software (Stoelting), and the average freezing time was calculated.

## Unconditioned innate fear response test

An unconditional innate fear response test was performed to evaluate innate fear, as previously described (Yun et al., 2019). Briefly, the mice were placed in a chamber in which 30 µl of 2,5-dihydro-2,4,5-trimethylthiazoline (TMT), a synthetic

component found in fox feces, was added. The video was recorded for 15 min, and TMT-induced freezing was analyzed using 6.36 software (Stoelting). The percentage of freezing time for every 3 min was plotted.

## Statistical analysis

All the statistical analyses were performed using GraphPad Prism 5 software (GraphPad Software, Inc., La Jolla, CA). The data are shown as the means ± SEMs. Statistical significance was determined using two-tailed unpaired Student's t test. For multiple comparisons, two-way ANOVA followed by the Bonferroni post hoc correction was performed. The criterion for statistical significance was set at a p value less than 0.05.

## Results

### FAM19A5-LacZ KI mice as a partial FAM19A5 KO mouse model

To generate a partial knockout FAM19A5 mouse model, we integrated LacZ upstream of exon 4 of the FAM19A5 gene. The insertion of the LacZ gene upstream of this exon led to the exclusion of two amino acids from the translated protein. Additionally, the disruption of normal splicing caused the translated mRNA to include an intronic sequence after the third exon. This intron introduces a stop codon after ten additional amino acids (Fig 1A). As a result, the secreted mutant (MT) FAM19A5 protein is 97 amino acids in length, compared to 89 amino acids in the wild-type (WT) protein.

We first investigated whether the insertion of the LacZ gene alters the mRNA levels of FAM19A5. Using qRT-PCR targeting exon 1 and exon 3, we measured the FAM19A5 mRNA levels in the cortex and hippocampus. The results revealed no significant differences in the FAM19A5 mRNA levels between the wild-type and LacZ KI mice in both brain regions (Fig 1B). We then examined FAM19A5 protein levels via Western blotting, which revealed a significant decrease in FAM19A5 protein levels in LacZ heterozygote mice, with an even greater reduction in LacZ homozygote mice (Figs 1C and D). Notably, Western blot analysis revealed the presence of the potential non-glycosylated MT FAM19A5, showing a slight size difference attributable to the eight additional amino acids at the C-terminal end (Fig 1C). To further confirm the reduced protein levels, we quantified the secreted levels of both glycosylated and non-glycosylated FAM19A5 in the cerebrospinal fluid (CSF) via ELISA [10]. Consistent with the Western blot results, FAM19A5 protein levels in the CSF of LacZ mice were significantly lower compared to WT mice. Specifically, heterozygous LacZ mice exhibited a 38.5% decrease in FAM19A5 levels compared to wild-type mice, while homozygous LacZ KI mice showed a 65.6% decrease (Fig 1E). FAM19A5 degradation pathways were evaluated *in vitro* using HEK293 cells. The cells were treated with cycloheximide, a protein synthesis inhibitor, either alone or in combination with MG132, a ubiquitin-proteasome pathway inhibitor. Both the WT and MT FAM19A5 proteins displayed similar degradation kinetics following cycloheximide treatment, and this degradation persisted despite MG132 co-treatment. This suggests that both WT and MT FAM19A5 may not be primarily degraded via the ubiquitin-proteasome pathway, and that they may utilize alternative pathways (S1 Fig). However, it is important to note that *in vitro* results may not fully reflect the degradation processes occurring *in vivo* in the brain.

It has been demonstrated that FAM19A5 specifically binds to the postsynaptic adhesion molecule LRRC4B rather than S1PR2 [7], thereby promoting synapse reduction [11]. To assess whether the MT FAM19A5 retains its ability to bind to LRRC4B, we performed co-immunoprecipitation experiments using HEK293 cells expressing LRRC4B and FAM19A5. Although MT FAM19A5 was expressed at lower levels than WT FAM19A5 in HEK293 cells, MT FAM19A5 still bound to LRRC4B similar to WT FAM19A5 (Fig 1F). As the relatively high affinity binding of MT FAM19A5 in the co-immunoprecipitation experiments can be attributed to an artifact of the overexpression system in HEK293 cells, we further quantitatively compared the binding affinity of MT FAM19A5 for LRRC4B with that of WT FAM19A5 via ELISA. MT FAM19A5 exhibited a 12.9-fold lower binding affinity for LRRC4B compared to WT FAM19A5 ($EC_{50}$ with MT vs. WT FAM19A5: 2564.0, 198.8 pM, Fig 1G).

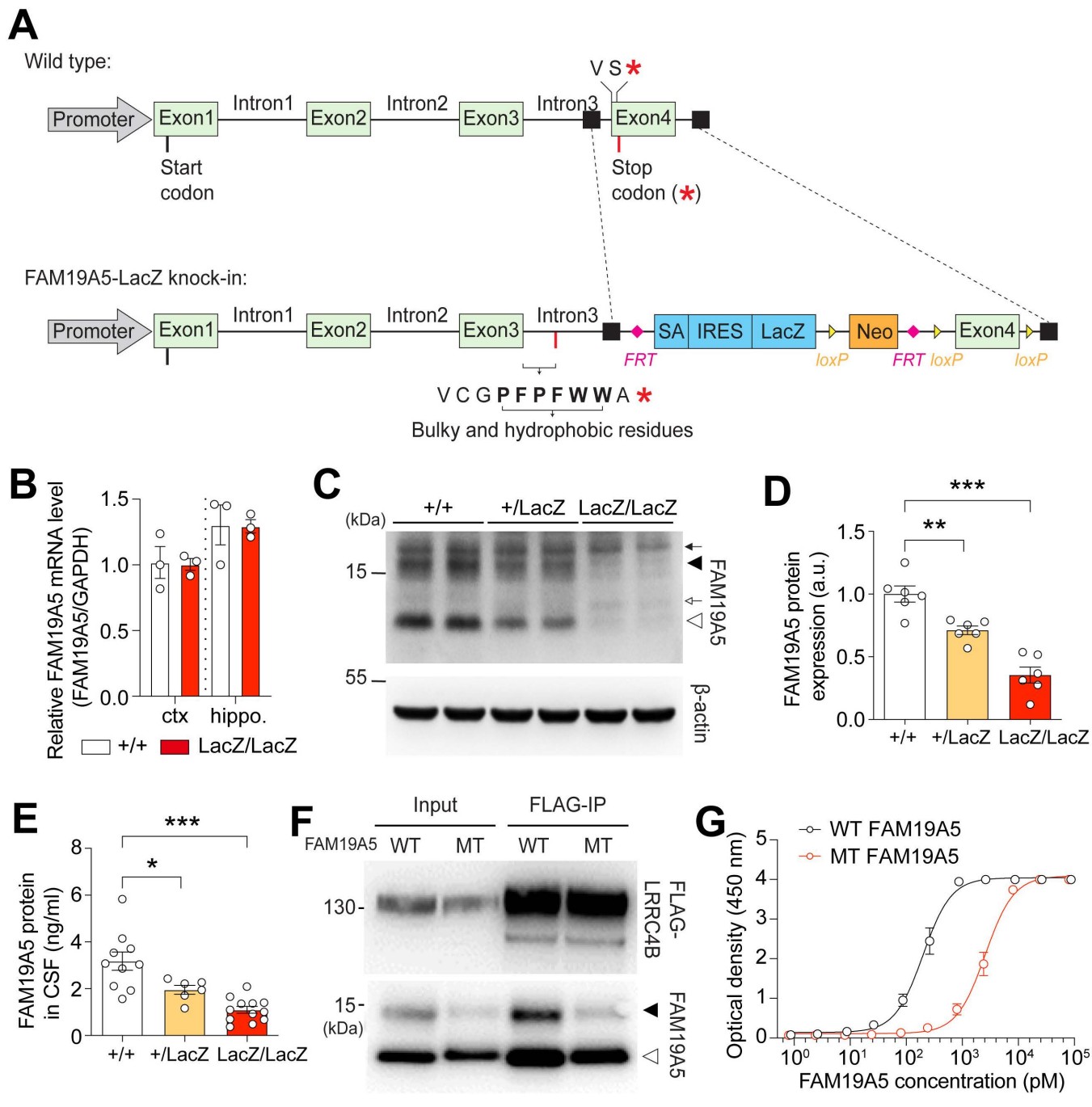

**Fig 1. Reduction of FAM19A5 protein levels in FAM19A5-LacZ KI mice.** (A) FAM19A5-LacZ KI construct. Additional 10 amino acids are added after exon 3 when mRNAs are translated from the FAM19A5-LacZ gene. (B) Quantification of FAM19A5 mRNA levels in the cortex and hippocampus of FAM19A5 LacZ mice. n = 3 each, unpaired T test, p = 0.9145 (cortex), 0.9552 (hippocampus). (C) Western blot analysis of FAM19A5 in whole brain lysates from FAM19A5-LacZ KI mice and WT littermates. The open and solid arrowheads indicate non-glycosylated and glycosylated WT FAM19A5, respectively. The open arrow represents a potential non-glycosylated MT FAM19A5 from FAM19A5$^{LacZ/LacZ}$ mice, and the solid arrow shows a non-specific band. (D) Quantification of combined glycosylated and non-glycosylated FAM19A5 protein expression levels in the brains of FAM19A5$^{+/LacZ}$, FAM19A5$^{LacZ/LacZ}$, and FAM19A5$^{+/+}$ littermates, n = 6, One way ANOVA followed by Tukey's multiple comparisons test. (E) Quantification of FAM19A5 protein level in the CSF of FAM19A5 LacZ mice. n = 6–12, One way ANOVA followed by Bonferroni's multiple comparisons test, *P = 0.0282, ***P < 0.0001. (F) Interactions between WT or MT FAM19A5 with LRRC4B were determined by co-immunoprecipitation. (G) The binding affinity of WT or MT FAM19A5 proteins to LRRC4B was measured via ELISA. n = 10 each. Data are presented as the mean ± SEM.

Overall, these findings indicate that the FAM19A5-LacZ KI model expresses reduced levels of the MT FAM19A5 with decreased binding affinity to LRRC4B, suggesting that this mouse model is suitable for studying FAM19A5 function *in vivo*.

### Reduced body weight and food intake in FAM19A5-LacZ KI mic

Since FAM19A5 expression begins as early as embryonic day 10.5 [3], a reduction in FAM19A5 expression throughout development in FAM19A5-LacZ KI mice might cause abnormalities. When heterozygous male and female mice were mated to generate homozygous mice, 19.92% (22.737% male and 17.054% female) of the FAM19A5^LacZ/LacZ mice were born, which was lower than the expected 25%. Importantly, these mice displayed similar sex ratios with no visible physical deformities and displayed normal postnatal growth and survival compared with their FAM19A5^+/+ littermates.

Previous studies using FAM19A5 KO mice reported decreased body weight [4]. To determine whether partial knock-out of FAM19A5 induced by LacZ KI would yield similar results, we compared body weight changes over time between FAM19A5 LacZ KI mice and wild-type mice. Despite normal growth and survival, FAM19A5-LacZ mice exhibited significantly lower weights compared to FAM19A5^+/+ mice (Figs 2A and B). This reduction appeared to be more pronounced in male mice. Our findings are consistent with previous research on FAM19A5 KO mice [4], suggesting that KI and KO mice share similar phenotypes in terms of body weight. We further investigated whether there were changes in food intake in FAM19A5 LacZ mice. Considering the more significant body weight variations in male mice, we observed changes in food intake only in male mice. We found that daily food intake was relatively lower in FAM19A5^LacZ/LacZ mice than in FAM19A5^+/+ mice, with significant decreases at the 54th, 56th, and 57th days of age (Fig 2C).

### Gross brain morphology in FAM19A5-LacZ KI mice

Given its early expression and predominant expression in neuronal stem cells (NSCs) and progenitor cells within the brain of germinal zones [3], a partial reduction in FAM19A5 may affect normal brain development and growth. To investigate this possibility, whole-brain size was measured in adult FAM19A5-LacZ KI mice and compared with WT mice. Equal numbers of male and female mice were used in this analysis. There were no significant differences in total brain length, width, and the cortical length between the FAM19A5-LacZ KI mice and their WT littermates (Figs 3A–C).

To assess potential effects on cortical development, we performed detailed histological measurements. The thicknesses of various cortical regions, including the motor, somatosensory, visual, and auditory cortex were measured in Nissl-stained coronal brain sections. FAM19A5-LacZ KI mice exhibited normal cortical thickness (Figs 3D–G). Notably, some FAM19A5-LacZ KI mice displayed enlarged lateral ventricles (LVs) compared to those of WT mice. Hence, the area

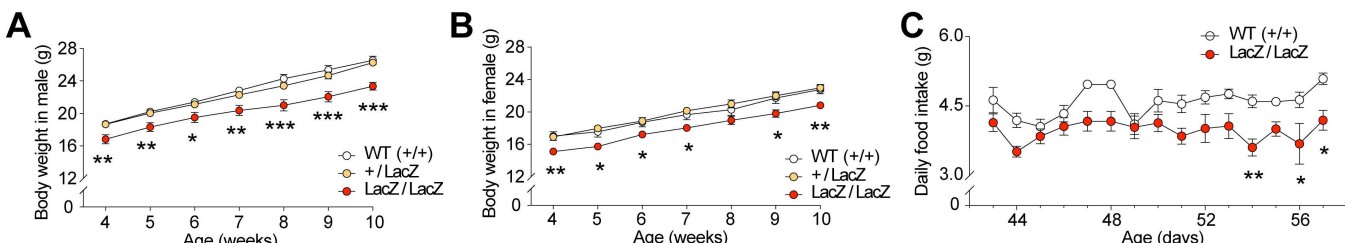

**Fig 2. Reduced body weight in FAM19A5-LacZ KI mice.** (A-B) Time-dependent body weight changes in male (A) and female (B) FAM19A5-LacZ KI mice compared to FAM19A5^+/+ littermates, male: n = 16 (WT^+/+), 27 (FAM19A5^+/LacZ), 15 (FAM19A5^LacZ/LacZ), female: n = 11 (WT^+/+), 18 (FAM19A5^+/LacZ), 11 (FAM19A5^LacZ/LacZ). Two-way ANOVA followed by Bonferroni's multiple comparisons test. (C) Reduced daily food intake in male FAM19A5^LacZ/LacZ and WT^+/+ mice for 15 days, n = 7 (WT^+/+), 6 (FAM19A5^LacZ/LacZ). Data are presented as the mean ± SEM. Two-way ANOVA followed by Bonferroni's multiple comparisons test, *P < 0.05; **P < 0.01 and ***P < 0.001 vs. WT^+/+.

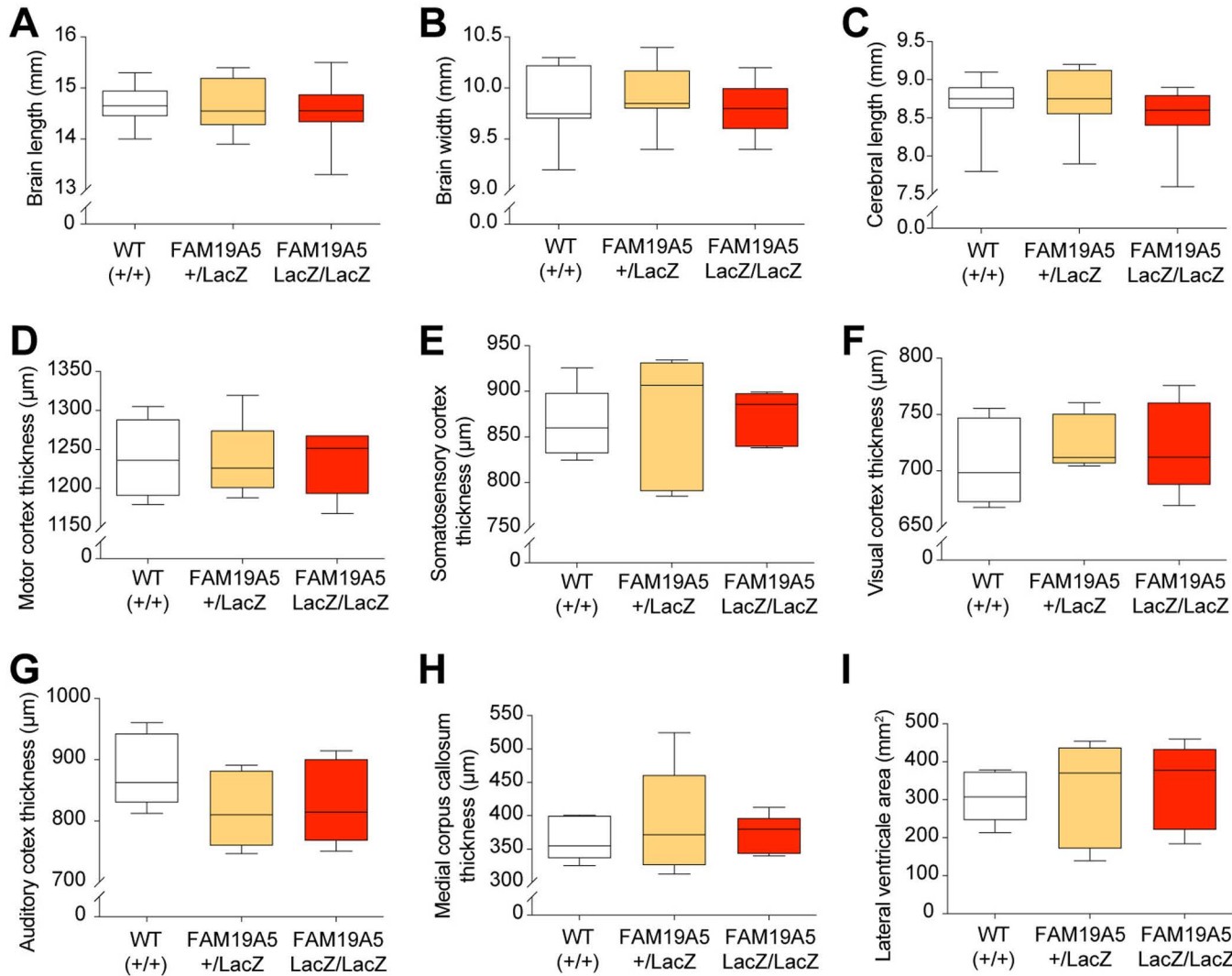

**Fig 3. Comparison of brain structure between FAM19A5-LacZ KI mice and their wild-type littermates.** (A) Whole brain length, (B) width, and (C) cerebral cortical length measurement between FAM19A5-LacZ KI mice and their FAM19A5[+/+] littermates. n = 14 (WT[+/+]), 14 (FAM19A5[+/LacZ]) and 16 (FAM19A5[LacZ/LacZ]). Thickness of the (D) motor, (E) somatosensory, (F) visual, and (G) auditory cortex between the mice (n = 5). Thickness of the (H) medial corpus callosum and (I) area of the lateral ventricle between the mice (n = 5 each). One-way ANOVA followed by Dunnett's multiple comparison test was used. Box plots visualize the data distribution, with the box representing the 50th percentile range (interquartile range) between the 25th and 75th percentiles within the minimum and maximum values.

of the LV and thickness of the corpus callosum were measured, but no significant differences were found compared with those of the WT group (Figs 3H and I).

Furthermore, we carefully investigated whether there were any differences in the thickness of the six layers comprising the motor, somatosensory, visual, and auditory cortices between FAM19A5 LacZ KI mice and WT mice. The six layers were clearly distinguished in Nissl-stained brain sections due to differences in neuronal density and soma size. Layer 1 exhibited a very low density of cells, primarily composed of dendrites and axons with only a few scattered neurons. Layers 2/3 contained small granule cells and medium-sized pyramidal neurons, with higher cell density in layer 2. Layer 4 is characterized by small, densely packed granule cells. Layer 5 is rich in large pyramidal cells. Layer 6 is composed of a mix

of cell types, including fusiform neurons and smaller pyramidal neurons. However, there were no significant differences in the thickness of the clearly defined layers between FAM19A5 LacZ KI mice and wild-type mice (S2 Fig). These histological analyses revealed normal gross brains in FAM19A5-LacZ KI mice, suggesting that a partial reduction in FAM19A5 may not significantly impact overall brain development.

## Dendritic spine analysis in FAM19A5-LacZ KI mice

Building on previous findings that complete FAM19A5 knockout reduces spine density [4], we hypothesized that a partial FAM19A5 KO mouse model may also exhibit a decrease in spine density. Spine density and length-to-width (LWR) were analyzed in both the basal and apical dendritic spines of pyramidal neurons from layers 2/3 and 5 of the motor and somatosensory cortices via Golgi staining. A balanced sex ratio was maintained for this analysis.

In the motor cortex, spine density was not significantly altered both in apical and basal dendrites of layers 2/3 and 5 between the FAM19A5$^{LacZ/LacZ}$ mice and their FAM19A5$^{+/+}$ littermates (Figs 4A, B, D, E). However, compared with their FAM19A5$^{+/+}$ littermates, FAM19A5$^{LacZ/LacZ}$ mice exhibited a significant increase in the LWR of both apical and basal dendrites in layer 5 pyramidal neurons (Fig 4F). The LWR of the apical and basal dendritic spines of layer 2/3 neurons was not significantly different between the FAM19A5$^{LacZ/LacZ}$ group and the FAM19A5$^{+/+}$ littermates (Fig 4C).

Similarly, in the somatosensory cortex, spine density remained unchanged between FAM19A5$^{LacZ/LacZ}$ and FAM19A5$^{+/+}$ mice (Figs 4G, H, J, K). However, FAM19A5$^{LacZ/LacZ}$ mice displayed a significant increase in the LWR of both apical and basal dendritic spines in layers 2/3 compared to FAM19A5$^{+/+}$ littermates (Fig 4I). The LWR of basal dendritic spines in layer 5 was not significantly altered in FAM19A5$^{LacZ/LacZ}$ mice (Fig 4L).

Morphologically, dendritic spines are categorized into various types. Filopodia and thin, characterized by their long, motile structure, are considered immature and unstable spine forms. In contrast, mature and stable spines, known as mushroom spines, are typically shorter and wider in size [14]. The observed increase in the LWR of spines in FAM19A5$^{LacZ/LacZ}$ mice suggested a potential increase in the number of immature dendritic spines in these adult mice.

To test this hypothesis, we classified the spines into three types: filopodia/thin, branched, and mushroom spines. In the motor cortex, the FAM19A5$^{LacZ/LacZ}$ mice displayed no significant alterations in filopodia/thin, branched and mushroom spines compared to those of the FAM19A5$^{+/+}$ controls (Figs 5A–D). In contrast, the somatosensory cortex exhibited a significant increase in filopodia/thin spines, and a decrease in mushroom spines in both the apical and basal dendrites of layer 2/3 neurons compared with those of the FAM19A5$^{+/+}$ controls (Figs 5E and F). In layer 5 of the somatosensory cortex, both apical and basal dendrites showed no significant differences in the spine densities between FAM19A5$^{LacZ/LacZ}$ and FAM19A5$^{+/+}$ mice (Figs 5G and H). The morphology of spines is closely associated with their function and synaptic strength [15,16]. The observed shift towards immature spine types in FAM19A5-deficient mice suggests an alteration in synaptic function, which could potentially influence the reduced body weight and daily food intake observed in these animals.

## General locomotion, exploratory and motor function in FAM19A5-LacZ KI mice

The observed shift towards immature spines and reduction of mature spines in FAM19A5-LacZ KI mice, which could indicate altered synaptic function, might disrupt normal brain function and contribute to behavioral abnormalities. To investigate this possibility, we performed a series of behavioral tests. To minimize the confounding effects of hormonal fluctuations, particularly from the estrous cycle in females, subsequent experiments were conducted exclusively with male mice. General locomotor, exploratory, and anxiety-related activities were observed via an open field test (OFT). The results showed a significant increase in locomotor activity in the FAM19A5$^{LacZ/LacZ}$ KI mice compared to that in the FAM19A5$^{+/+}$ KI mice, as indicated by the total distance traveled and the mean speed of movement (Figs 6A–C). This hyperactivity was primarily observed in the peripheral region of the open field but not in the central zone (Figs 6D–H). Notably, the time spent in the center zone did not significantly differ between FAM19A5-LacZ KI mice and WT littermates (Fig 6I).

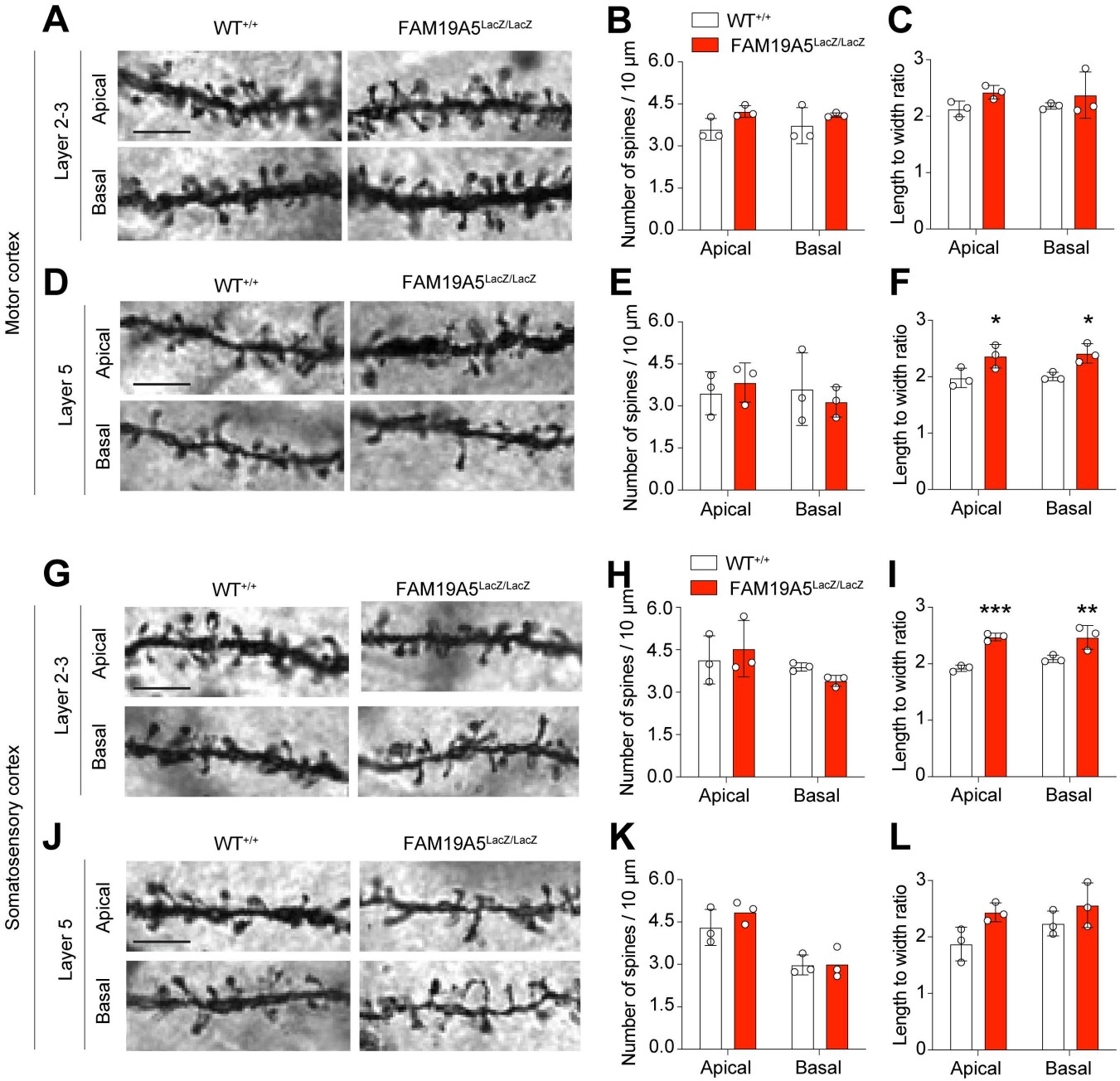

**Fig 4. Dendritic spine analysis in the adult FAM19A5-LacZ KI mouse brain.** (A) Golgi staining image of dendritic spines in layer 2/3 of the motor cortex. (B-C) And its quantification in terms of (B) density and (C) the length-width ratio. (D) Golgi staining image of dendritic spines in layer 5 of the motor cortex. (E-f) And its quantification in terms of (E) density and (F) the length-width ratio. (G) Golgi staining image of dendritic spines in layer 2/3 of the somatosensory cortex. (H-I) And its quantification in terms of (H) density and (I) the length-width ratio. (J) Golgi staining image of dendritic spines in layer 5 of the motor cortex. (K-L) And its quantification in terms of (K) density and (L) the length-width ratio. Data are presented as the mean ± SEM. n = 3 each, Two-way ANOVA followed by Bonferroni's multiple comparisons test, *P < 0.05, **P < 0.01 and ***P < 0.001 vs. WT +/+. Scale bars 5 μm.

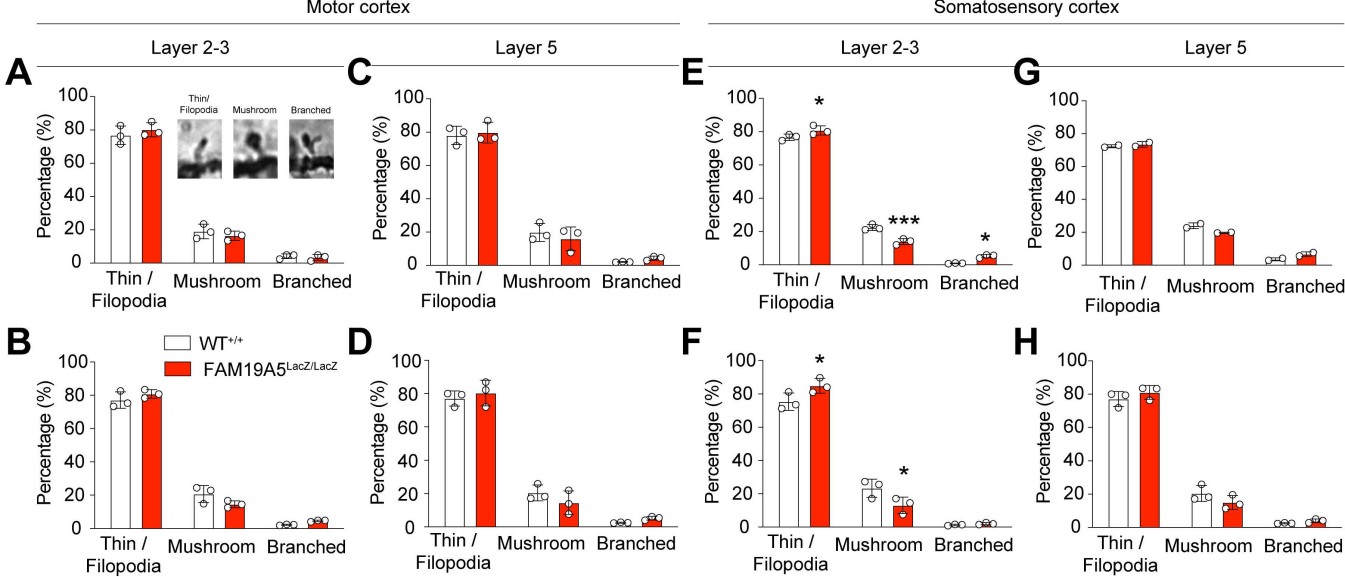

**Fig 5. Alterations in the dendritic spine type in the adult FAM19A5-LacZ KI mouse brain.** (A-B) Classified spine types in the apical (A) and basal (B) dendrites of layer 2/3 pyramidal neurons in the motor cortex. A representative image of thin/filopodia, mushroom, and branched spine was inserted in (A). (C-D) Classification of spine types in the apical (C) and basal (D) dendrites of layer 5 pyramidal neurons in the motor cortex. (E-F) Classified spine types in the apical (E) and basal (F) dendrites of layer 2/3 pyramidal neurons in the motor cortex. (G-H) Classification of spine types in the apical (G) and basal (H) dendrites of layer 5 pyramidal neurons in the motor cortex. Data are presented as the mean ± SEM. n = 3 (WT), 2 (FAM19A5$^{+/LacZ}$) and n = 3 (FAM19A5$^{LacZ/LacZ}$), *P < 0.05, ***P < 0.001 vs. WT.

Rearing, where mice stand upright on their hind limbs, is considered a measure of exploratory behavior. There are two types of rearing: supported rearing, where mice use the arena wall for balance, which is indicative of enhanced locomotor activity, and unsupported rearing, which reflects an emotional state [17,18]. Although there was no significant difference in total rearing number (S3A Fig), FAM19A5$^{LacZ/LacZ}$ KI mice displayed a significant increase in supported rearing accompanied by a decrease in unsupported rearing compared to that of the FAM19A5$^{+/+}$ group (S3B and S3C Figs). These findings suggest potential dampening of the emotional component of exploration in FAM19A5-LacZ KI mice. This shift, with increased supported rearing and decreased unsupported rearing, might indicate reduced anxiety-like behavior in these mice compared with their wild-type littermates.

In addition to rearing, grooming was analyzed to assess repetitive, compulsive-like behavior. There was no significant difference in total grooming time between FAM19A5$^{LacZ/LacZ}$ KI and FAM19A5$^{+/+}$ mice, suggesting that FAM19A5 deficiency does not lead to alterations in repetitive grooming behavior (S3D Fig).

Next, we performed the rotarod test and hanging wire test to evaluate motor learning, coordination, and strength in FAM19A5-LacZ KI mice. In the rotarod test, both FAM19A5$^{LacZ/LacZ}$ and FAM19A5$^{+/+}$ control mice showed similar level of falling latency (S3E Fig). In the hanging wire test, there was no significant difference in the number of falls between the two groups (S3F Fig). Overall, these results suggest normal motor function in FAM19A5-LacZ KI mice.

## Repetitive and compulsive-like behavior in FAM19A5-LacZ KI mice

Most neurodevelopmental disorders characterized by hyperactivity, such as attention-deficit/hyperactivity disorder (ADHD) and autism spectrum disorder (ASD), are accompanied by repetitive or compulsive-like behavior. In the OFT, FAM19A5-LacZ KI mice tended to exhibit reduced grooming time, suggesting an absence of repetitive-like behavior. To further investigate repetitive behavior in hyperactive FAM19A5-LacZ KI mice, the marble burying test and nestlet shredding test

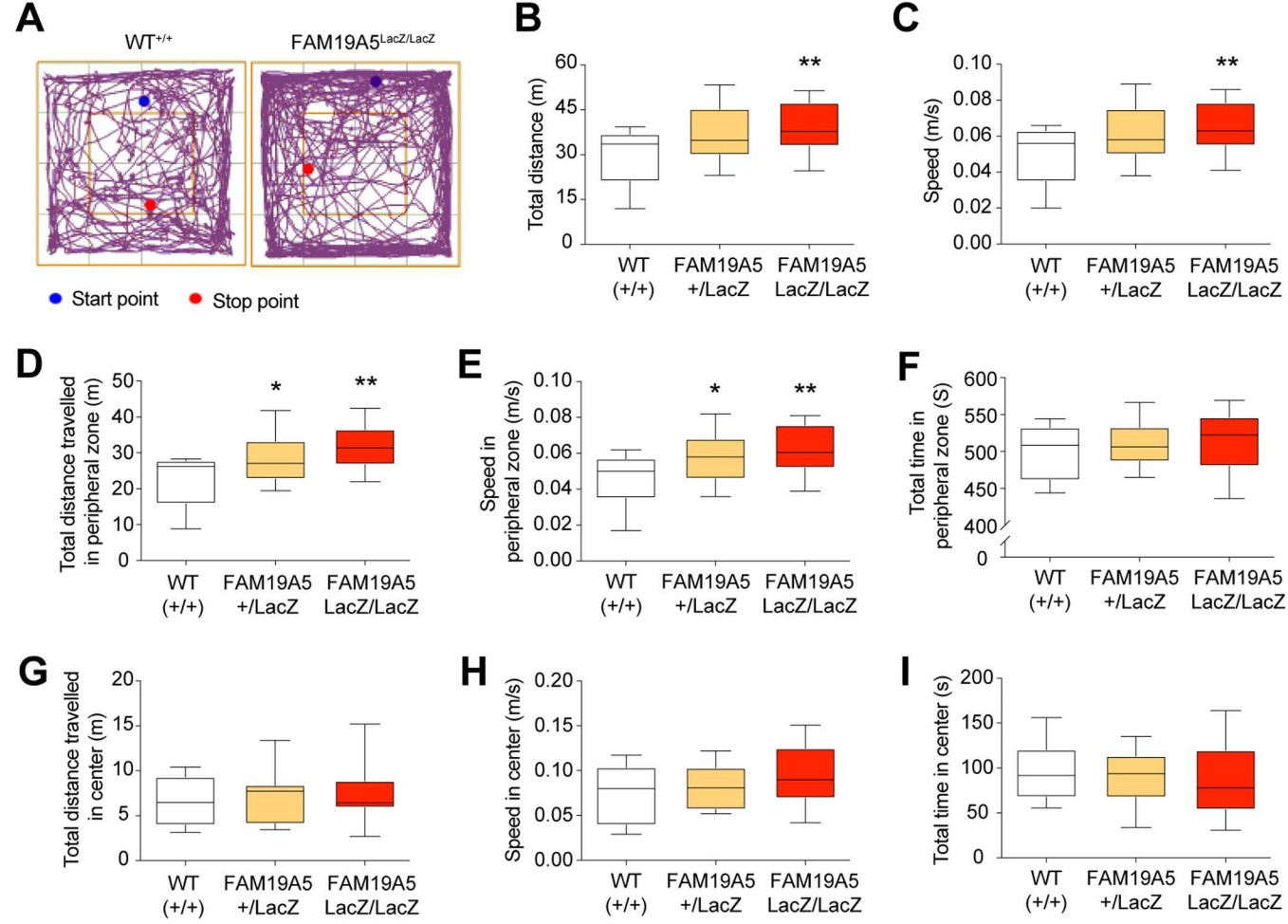

**Fig 6. Hyperactivity in FAM19A5-LacZ KI mice.** (A) Representative track plot of movements in the open field box during 10 min of exploration in FAM19A5LacZ/LacZ mice and their FAM19A5+/+ littermates. (B) Total travel distance and (C) mean speed of movement in the open field, respectively. (D) Total travel distance, (E) mean speed of movement and (F) time spent in the peripheral zone of the open field box, respectively. (G) Total travel distance, (H) mean speed movement and (I) time spent in the center region of the open field, respectively. n = 13 (WT+/+), n = 11 (FAM19A5+/LacZ), n = 16 (FAM19A5LacZ/LacZ). Box plots visualize the data distribution, with the box representing the 50th percentile range (interquartile range) between the 25th and 75th percentiles within the minimum and maximum values. *P < 0.05 **P < 0.01 vs. FAM19A5+/+.

were performed [19]. FAM19A5LacZ/LacZ mice showed a significant decrease in the number of buried marbles compared to that of FAM19A5+/+ mice (S3G Fig), suggesting reduced repetitive behavior in the FAM19A5-deficient mice. Although no significant difference was detected in the nestlet shredding test (S3H Fig), these findings collectively indicate the absence of repetitive or compulsive-like behavior in FAM19A5-LacZ KI mice.

## Anxiety/depressive-like behavior in FAM19A5-LacZ KI mice

To further confirm the anxiety-related behavior of FAM19A5-LacZ KI mice, an elevated plus maze test (EPMT) was performed. Compared with control mice, FAM19A5LacZ/LacZ mice showed significant increases in distance traveled and speed of movement in the maze (S4A–C Figs). Although the total time spent and total number of entries in the open arms of the maze were greater in the FAM19A5LacZ/LacZ mice than in the FAM19A5+/+mice, the differences were not statistically significant (S4D and S4E Figs).

The tail suspension test (TST) was performed to assess depressive-like behavior in FAM19A5-LacZ KI mice. As shown in S4F Fig, immobility did not significantly differ between the FAM19A5-LacZ KI and WT groups, indicating the absence of depressive-like behavior in FAM19A5-LacZ KI mice.

### Learning and memory function in FAM19A5-LacZ KI mice

To investigate whether the increase in immature spines in FAM19A5-LacZ KI mice leads to impaired learning and memory function, we performed the Y-maze test for spatial working memory and the novel object recognition (NOR) test for recognition memory, respectively. In the Y-maze test, no significant difference in spontaneous alterations was observed between FAM19A5-LacZ KI and WT mice (S5A Fig). This finding suggested normal spatial working memory in the FAM19A5-deficient mice. Consistent with the OFT and EPMT results, FAM19A5-LacZ KI mice traveled a significantly greater distance and moved at a faster speed within the maze (S5B and S5C Figs).

In the NOR test, there were no significant differences in preference for the novel object after 6 h (short-term memory) (S5D Fig) or after 24 h (long-term memory) (S5E Fig) between FAM19A5-LacZ KI and WT mice. These results indicated normal recognition memory, both short-term and long-term, in FAM19A5-LacZ KI mice.

### Fear response memory and the innate fear response in FAM19A5-LacZ KI mice

We then investigated whether FAM19A5-deficient mice have deficits in fear response memory consolidation. To investigate this, a Pavlovian fear conditioning test was performed. FAM19A5$^{LacZ/LacZ}$ mice exhibited significantly less freezing behavior than their FAM19A5$^{+/+}$ littermates did during the initial fear acquisition period (Fig 7A). While all the mice displayed increased freezing on the conditioning day, compared with their FAM19A5$^{+/+}$ littermates, the FAM19A5$^{LacZ/LacZ}$ mice failed to show robust freezing behavior indicative of fear consolidation. Consequently, FAM19A5-LacZ KI mice exhibited a significantly reduced freezing rate during both the contextual (Fig 7B) and auditory fear memory tests (Fig 7C). These findings suggest potential deficits in fear response memory consolidation in FAM19A5-LacZ KI mice.

The reduction in the freezing rate during the fear acquisition phase in the FAM19A5$^{LacZ/LacZ}$ group might be due to their hyperactivity. To investigate whether impaired fear acquisition is related to a general deficit in the fear response, the innate fear response was tested via threat match tracking (TMT). Typically, when mice encounter predators, they tend to exhibit instinctive behaviors such as freezing as a measure of fear.

Upon exposure to the TMT, FAM19A5$^{LacZ/LacZ}$ mice exhibited a significant decrease in freezing time compared to FAM19A5$^{+/+}$ mice. These mice displayed a significant decrease in freezing time throughout the initial 6 mins of TMT exposure. However, their freezing behavior gradually increased over time, eventually reaching levels similar to those of their FAM19A5$^{+/+}$ littermates (Fig 7D). This, coupled with their hyperactivity, may contribute to the impaired fear memory consolidation observed in these animals.

### Discussion

In this study, FAM19A5-LacZ KI mice were used as a partial FAM19A5 KO mouse model to investigate loss-of-function studies of FAM19A5. Western blot analysis confirmed a partial reduction in the FAM19A5 protein in the adult brains of FAM19A5-LacZ KI mice compared with WT mice. General physical observation revealed no visible differences in gross appearance and survival among the FAM19A5-LacZ KI mice. However, both male and female FAM19A5$^{LacZ/LacZ}$ mice had low body weights, which is consistent with the findings of a recent study using a complete FAM19A5 KO mouse model [4]. Furthermore, compared with the FAM19A5$^{+/+}$ mice, the FAM19A5$^{LacZ/LacZ}$ mice consumed relatively less food. Additionally, a study reported that alterations in FAM19A5 expression in the brain depend on nutritional status [20]. They observed increased FAM19A5 mRNA levels in the hypothalamus, cortex, and hippocampus of fasted mice. Notably, the hypothalamus is a brain region known to regulate feeding and satiety and body metabolism [21]. Moreover, X-gal staining of adult mouse brains revealed widespread FAM19A5 expression in the hypothalamus, which was further confirmed by single-cell

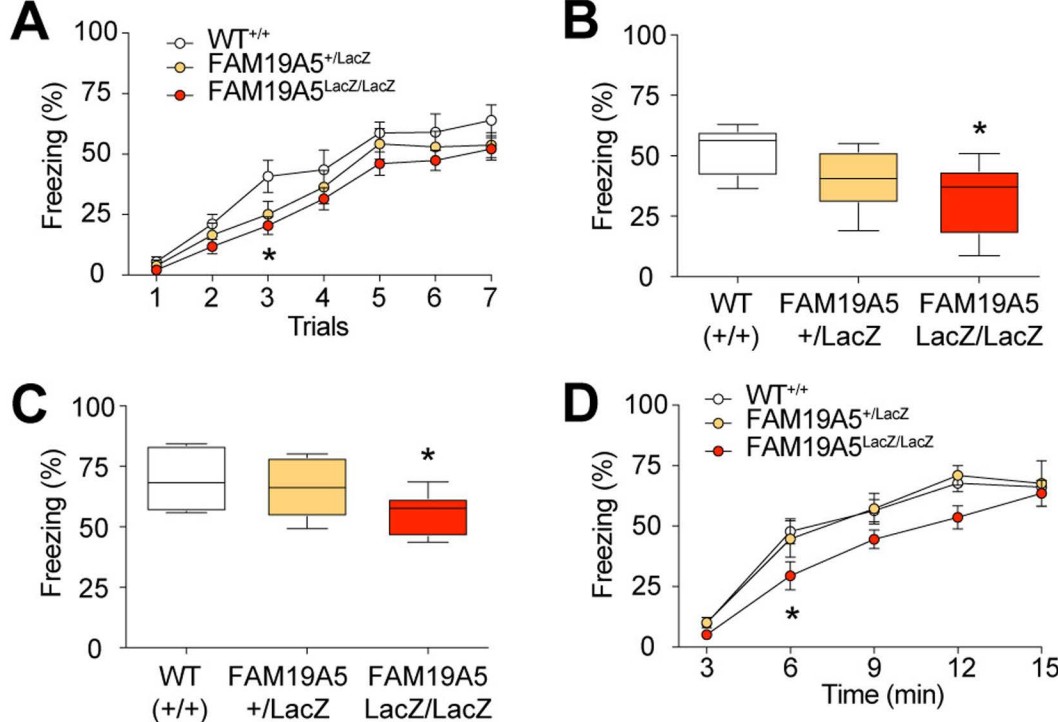

**Fig 7. Fear response in FAM19A5-LacZ KI mice.** (A) Freezing rates were measured in individual trials during the acquisition session in FAM19A5-LacZ KI mice and their littermates. The session consisted of 7 repetitive tones (5 kHz, 70 dB, 30 s) followed by a foot shock (0.7 mA, 2 s). (B-C) Freezing rate during the (B) contextual fear memory test and (C) auditory fear memory test. n = 7 (WT$^{+/+}$), n = 7 (FAM19A5$^{+/LacZ}$) and n = 7 (FAM19A5$^{LacZ/LacZ}$). (D) The freezing rate in response to 2,5-dihydro-2,4,5-trimethylthiazoline exposure was measured in FAM19A5-LacZ KI mice and their littermates to test the innate fear response. n = 15 (WT$^{+/+}$), 4 (FAM19A5$^{+/LacZ}$) and 11 (FAM19A5$^{LacZ/LacZ}$). Data are presented as the mean ± SEM. Box plots visualize the data distribution, with the box representing the 50th percentile range (interquartile range) between the 25th and 75th percentiles within the minimum and maximum values *P < 0.05 vs. WT$^{+/+}$.

RNA sequencing results from www.mousebrain.org showing high FAM19A5 expression in the NPY/AgRP neurons of the arcuate hypothalamic nucleus. These neurons are known as orexigenic neurons, the activation of which results in the induction of food intake [22]. Therefore, the partial reduction in FAM19A5 in FAM19A5-LacZ KI mice likely contributes to their lower body weight through a combination of decreased food intake and potentially increased energy expenditure due to their hyperactive behavior.

FAM19A5-LacZ KI mice exhibited a normal brain size, as assessed by gross observation. Histological analysis of adult FAM19A5-LacZ KI brain sections revealed no discernible abnormalities in gross brain structures compared with those in WT brains. Measurement of cortical layer thickness in various regions, including the motor, somatosensory, visual, and auditory cortex, revealed no significant differences between FAM19A5-LacZ KI and WT mice. However, complete deletion of FAM19A5 resulted in reduced brain weight and smaller brain structure [4], suggesting its crucial role in early brain development. This notion is further supported by a recent study demonstrating that FAM19A5 is predominantly expressed in germinal zones such as the ventricular zone and ganglionic eminence during the embryonic developmental stage [3]. These regions are pools of neural stem cells where division and differentiation into various types of cells occur. Therefore, FAM19A5 likely plays a role in early brain development, and partial knockout of FAM19A5 may have less of an impact on this process.

Furthermore, analysis of dendritic spine morphology in the motor and somatosensory cortices of FAM19A5-LacZ KI mice revealed an increase in the proportion of longer and thinner spines, indicative of a shift toward a more immature spine population. The spine density was not significantly altered in any of the analyzed brain regions. Therefore, the results of this study using FAM19A5-LacZ KI mice differ significantly from those of Huang et al., who used FAM19A5-LacZ KO mice, particularly regarding alterations in spine density and the proportions of immature and mature spine populations [4]. Although the brain regions analyzed in the two studies are different, these contrasting results lead to different interpretations of the function of FAM19A5, warranting a more detailed discussion.

Huang et al. reported that the spine composition in the CA1 region of the hippocampus in both FAM19A5 KO and wild-type mice showed an excessive proportion of mushroom spines compared to thin spines. Additionally, all spine types were reduced by more than 50% in the KO mice, suggesting that FAM19A5 may play a positive role in spine formation. However, according to other researchers, the spine composition in CA1 typically shows that mushroom spines are much less frequent than thin spines [11,23]. Since thin and mushroom spines represent immature and mature spines, respectively, this ratio is critical.

Our results revealed that the proportion of mushroom spines was lower than that of thin spines, aligning with general findings reported by other researchers [11,23]. Interestingly, in the KI mice, there was a slight increase in the proportion of thin spines but a decrease in mushroom spines compared to the wild-type mice, without significant changes in total spine density. Therefore, our KI mouse results suggest that FAM19A5 deficiency may not alter spine density but might be involved in the elimination of immature spines. For instance, improper pruning of immature spines in KI mice could lead to a relative decrease in mushroom spines, potentially leading to hyperactivity [24].

This inconsistency in spine density and distribution may be attributable to the complete versus partial deficiency of the FAM19A5 protein in KO and KI mice, respectively. Since FAM19A5 is expressed from the early stages of brain development [3], the absence of FAM19A5 during early development may significantly affect adult brain structure, while partial deficiency may have a lesser impact on developmental defects. Additionally, these discrepancies may arise from differences in methodology, such as Golgi staining in this study versus the sparse labeling method following infection with an EGFP-containing AAV virus in the study of Huang et al. [4]. Technical limitations, including the analysis of low-resolution spine images and ambiguous classification criteria, could also contribute to these differences.

FAM19A5-LacZ KI mice displayed hyperactivity, which is in accordance with the findings of a study by Huang et al. [4]. Hyperactivity is a characteristic feature of the neurodevelopmental disorder ADHD. There are few human cases in which patients with developmental defects are reported to have defects in chromosome 22, where FAM19A5 is located [5,6]. Specifically, a microdeletion of chromosome 22, del22q13.32-q13.33, encompasses exon 4 of FAM19A5 and is associated with clinical features such as hyperexcitability and aggression, similar to those observed in ADHD. These findings suggest a potential role for FAM19A5 in neurodevelopment disorders. However, it is important to note that the microdeletion deletes not only the FAM19A5 gene but also other neighboring genes, including SHANK3, which has been extensively studied in relation to neurodevelopmental disorders [25,26].

FAM19A5-LacZ KI mice showed increased activity, primarily in the peripheral region of the open field, with a greater tendency toward rearing behavior. Notably, the mice exhibited more supported rearing and consequently less unsupported rearing. Supported rearing is primarily associated with locomotor activity, whereas unsupported rearing is an exploratory behavior dependent on the hippocampus and can be indicative of a mouse's emotional state [18]. This pattern suggested enhanced locomotor activity in FAM19A5-LacZ KI mice.

Previous studies have demonstrated that FAM19A5 is prominently expressed in layers 2/3 and 5 of the cortex and the CA regions of the hippocampus [3,9]. FAM19A5 LacZ mice exhibited hyperactivity and deficits in fear memory consolidation, suggesting a crucial role for FAM19A5 in these behavioral and cognitive processes. The hippocampus is well-established as a key region for memory consolidation and is implicated in hyperactivity. The somatosensory cortex also plays a significant role in memory consolidation, particularly motor memory, through mechanisms such as up- and

downstates during sleep and learning-related plasticity [27,28]. Additionally, it is associated with hyperactive behavior through layer-specific mechanisms [29]. In our study, the observed significant increase in immature spines within somato-sensory cortical layers 2/3 and 5, rather than within the motor cortex, supports the hypothesis that FAM19A5 is essential for synaptic refinement in these specific brain regions. These findings collectively indicate that FAM19A5 is likely a key modulator of neural circuit development and function, with implications for higher-order cognitive processes and behavioral control.

Furthermore, the FAM19A5-LacZ KI mice tended to exhibit less grooming behavior. Grooming is a measure of repetitive or compulsive-like behavior that occurs in individuals with neurodevelopmental disorders such as ASD [30]. Similarly, FAM19A5-LacZ KI mice displayed reduced marble burying behavior, a common test for repetitive actions [31]. This suggests a lack of repetitive behavior despite hyperactivity. The decrease in buried marble may be due to FAM19A5-LacZ KI mice spending more time in other activities, such as supported rearing, leaving less time for digging.

Further behavioral analysis revealed the absence of anxiety/depressive-like behavior, normal spatial working memory, and normal recognition memory in FAM19A5-LacZ KI mice compared with their WT littermates. However, a study on the FAM19A5 KO model by Huang et al. showed the presence of depressive-like behavior and impaired spatial memory [4]. The dynamic expression pattern of FAM19A5, with its potential role in early development [3], suggested that complete knockout from conception might have a more profound impact than the partial reduction observed in our FAM19A5-LacZ KI model. This could explain the abnormal neurogenesis observed by Huang et al., which might in turn contribute to the learning and memory impairments reported in their study.

FAM19A5 is highly expressed in rodent brain regions mediating fear learning and memory, including the amygdala and hippocampus [32]. FAM19A5-LacZ KI mice displayed impaired fear acquisition during the fear conditioning period. Consequently, analyses of both context-induced fear memory and tone-induced fear memory in mice were limited. The impaired fear acquisition in the mice might be due to their hyperactivity, but it might also be related to deficits in the innate fear response or sensory functions. Since all the mice displayed similar vocalization and jumping behavior following unconditioned and conditioned stimuli during the fear acquisition period, their sensory function was likely intact. The unconditioned innate fear response test revealed a delayed response to TMT in FAM19A5-LacZ KI mice compared with WT mice. This finding suggested that the impaired fear acquisition in FAM19A5-LacZ KI mice may be related to a general deficit in their innate fear response.

Understanding the function of FAM19A5 in the brain is crucial. Recent clinical studies revealed its genetic association with brain development-related symptoms such as ADHD and autism [6], as well as neurodegenerative diseases, such as Alzheimer's disease [33,34]. In this study, mice with partial FAM19A5 reduction exhibited reduced body weight, abnormalities in dendritic spine maturation and structure, and hyperactive behavior. These findings strongly suggest that FAM19A5 plays important roles in spine regulation, synaptic transmission regulation, and normal brain development. Further supporting this notion, a recent study elucidated the potential role of FAM19A5 in regulating synaptic elimination. Overexpression of FAM19A5 in cultured hippocampal neurons significantly reduced spine density. Similarly, treatment with the FAM19A5 protein led to a comparable reduction, while neutralizing FAM19A5 with an anti-FAM19A5 antibody restored spine density [11]. These findings suggest that FAM19A5 may play a role in maintaining synaptic plasticity by balancing synapse formation and elimination.

FAM19A1-4 interact with neurexin, a presynaptic adhesion protein crucial for diverse neurological processes [35]. However, FAM19A5 is proposed to function as a ligand for other synaptic receptors. A recent study revealed that FAM19A5 is a synaptolytic factor, mediating the removal of synapses through binding to LRRC4B, a postsynaptic adhesion molecule [11]. This process may play a role in removing superfluous synapses and facilitating selective synapse maturation. In FAM19A5 LacZ mice, we observed an elevated number of immature spines, suggesting a defect in synaptic pruning in the absence of FAM19A5. This phenotype may arise from a failure in the focused maturation of selected spines, leading to an accumulation of immature synaptic structures [24].

In conclusion, this study employed FAM19A5-LacZ KI mice as a model for partial FAM19A5 deficiency. Western blot analysis confirmed a reduction in FAM19A5 protein levels in the adult brains of these mice compared to wild type controls. Interestingly, FAM19A5-LacZ KI mice displayed lower body weights and reduced food intake, potentially linking FAM19A5 to feeding behavior regulation in the hypothalamus. While gross brain structure appeared normal, further investigation into specific neuronal populations and their activity is warranted. Overall, these findings suggest a role for FAM19A5 in body weight regulation and potentially early brain development, highlighting the need for further exploration of its function in the nervous system.

## Supporting information

**S1 Fig. Time-dependent degradation of the FAM19A5 protein in HEK293 cells.** (A) Western blot analysis of the time-dependent degradation of the wild-type (WT) and mutant (MT) FAM19A5 proteins, and beta-actin. Changes in protein levels were observed at various time points after co-treating cells with either cycloheximide (CHX) alone or CHX and MG132 simultaneously. (B, C) Quantification of the (B) FAM19A5 protein bands (n = 3 each, comparison of linear regression models, slope: $F_{(3, 40)} = 0.1966$, $P = 0.8981$), and (C) beta-actin (n = 6 each, Two way ANOVA followed by Šídák's multiple comparisons test, **$p = 0.0066$).
(DOCX)

**S2 Fig. Characterization of cortical layers in FAM19A5 LacZ KI mice.** (A) Nissl-stained brain slices. The motor, somatosensory, visual, and auditory cortical regions are indicated by the black box. Scale bar, 1 mm. (B) Layers 1–6 of the motor, somatosensory, visual, and auditory cortex in WT, heterozygous, and homozygous LacZ KI mice. Scale bar, 50 μm. (C) Quantification of cortical layer thickness (n = 5). Data are presented as mean ± SEM.
(DOCX)

**S3 Fig. Rearing and grooming behavior in FAM19A5-LacZ KI mice.** (A) Total number of rearings, (B) supported rearing, (C) unsupported rearing, and (D) total grooming time during the 10 min of exploration period in the OFT. FAM19A5$^{+/+}$, n = 13 and FAM19A5$^{LacZ/LacZ}$, n = 15. Data are presented as the mean ± SEM. **$P < 0.01$ vs. FAM19A5$^{+/+}$. (E) Latency to fall in a rotating rod during 3 trials per day for 5 consecutive days (FAM19A5$^{+/+}$, n = 9 and FAM19A5$^{LacZ/LacZ}$, n = 7). (F) Latency to fall in a hanging wire test in 300 s long 3 consecutive trials with at least 15 min intertrial interval (FAM19A5$^{+/+}$, n = 18 and FAM19A5$^{LacZ/LacZ}$, n = 16). (G) Number of marbles buried during the 30 min exploration time (FAM19A5$^{+/+}$, n = 14; FAM19A5$^{+/LacZ}$, n = 9; FAM19A5$^{LacZ/LacZ}$, n = 14). (H) Percentage of shredded nestlet during 30 min of observation time (FAM19A5$^{+/+}$, n = 16; FAM19A5$^{+/LacZ}$, n = 13; FAM19A5$^{LacZ/LacZ}$, n = 15). Data are presented as the mean ± SEM. ***$P < 0.001$ vs. FAM19A5$^{+/+}$.
(DOCX)

**S4 Fig. The absence of anxiety and depressive-like behavior in FAM19A5-LacZ KI mice.** (A) Representative track plot of movements in the elevated plus maze during 15 min of exploration time in FAM19A5$^{LacZ/LacZ}$ and FAM19A5$^{+/+}$ littermates. (B and C) Total distance traveled and mean speed of movement in the elevated plus maze, respectively. (D and E) Percentage of time spent and total number of entries into the open arms of the maze, respectively. (F) Percentage of immobility time during the 5 min long TST. FAM19A5$^{+/+}$, n = 16; FAM19A5$^{+/LacZ}$, n = 12 and FAM19A5$^{LacZ/LacZ}$, n = 16. Data are presented as the mean ± SEM. *$P < 0.05$ and **$P < 0.01$ vs. FAM19A5$^{+/+}$.
(DOCX)

**S5 Fig. Normal learning and memory function in FAM19A5-LacZ KI mice.** (A-C) Total distance traveled, speed of motion and percentage of spontaneous alteration in Y-maze arms during 5 min of exploration period in FAM19A5$^{+/LacZ}$, n = 12; FAM19A5$^{LacZ/LacZ}$, n = 16 and FAM19A5$^{+/+}$, n = 16. (D) Percentage of preference to novel object during 10 min exploratory time in NOR test after 6 h for short term memory and (E) after 24 h for long term memory. FAM19A5$^{+/+}$,

n = 7; FAM19A5$^{+/LacZ}$, n = 3; FAM19A5$^{LacZ/LacZ}$, n = 8 for short term memory and FAM19A5$^{+/+}$, n = 5; FAM19A5$^{+/LacZ}$, n = 4; FAM19A5$^{LacZ/LacZ}$, n = 7 for long term memory. Data are presented as the mean ± SEM. *P < 0.05 vs. FAM19A5$^{+/+}$.
(DOCX)

**S6 Raw gel images.** Western blot original image in Figs 1C and 1F, S1A.
(PDF)

## Author contributions

**Conceptualization:** Anu Shahapal, Sangjin Yoo, Jong-Ik Hwang, Jae Young Seong.

**Data curation:** Anu Shahapal, Sumi Park, Sangjin Yoo, Jae Young Seong.

**Formal analysis:** Anu Shahapal, Sumi Park.

**Funding acquisition:** Jae Young Seong.

**Investigation:** Anu Shahapal, Sumi Park, Sangjin Yoo, Shi-Xun Ma, Jongha Lee, Hoyun Kwak, Jong-Ik Hwang, Jae Young Seong.

**Methodology:** Sumi Park, Jongha Lee, Hoyun Kwak.

**Project administration:** Jae Young Seong.

**Resources:** Jong-Ik Hwang, Jae Young Seong.

**Supervision:** Jae Young Seong.

**Visualization:** Anu Shahapal, Sangjin Yoo.

**Writing – original draft:** Anu Shahapal, Sangjin Yoo, Jae Young Seong.

**Writing – review & editing:** Sangjin Yoo, Jae Young Seong.

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
