## [Decision Letter · Decision Letter 0]

PONE-D-24-51742Partial FAM19A5 deficiency in mice leads to disrupted spine maturation, hyperactivity, and an altered fear responsePLOS ONE

Dear Dr. Seong,

Thank you for submitting your manuscript to PLOS ONE. After careful consideration, we feel that it has merit but does not fully meet PLOS ONE’s publication criteria as it currently stands. Therefore, we invite you to submit a revised version of the manuscript that addresses the points raised during the review process.

Additional comments :In Figure 5� the authors made observations on the difference of different types of dendritic spines between the wildtype and the mutant mice. To help the readers to understand the figure, the authors should provide an image illustrating the different type of spines such as filopodia, mushroom and branched spines with clear labeling. The submitted figures were shown in an order from Fig7 to Fig1, which is not the usual order for manuscript submission. Please re-order the figure sequence from Fig1 to Fig7 during the revision.

We look forward to receiving your revised manuscript.

Kind regards,

Hualin Fu

Academic Editor

PLOS ONE

“This research was supported by grants from Neuracle Science Co., Ltd and the Research Program of the National Research Foundation of Korea (NRF‐2020M3E5D9080794), which is funded by the Korea Government (MSIT).”

“S.A, SM.P, and JY.S are shareholders of Neuracle Science Co., Ltd. SJ.Y, HY.K and JH.L are employees of Neuracle Science.”

Reviewers' comments:

Reviewer's Responses to Questions

**Comments to the Author**

1. Is the manuscript technically sound, and do the data support the conclusions?

Reviewer #1: Partly

Reviewer #2: Yes

2. Has the statistical analysis been performed appropriately and rigorously? 

Reviewer #1: I Don't Know

Reviewer #2: Yes

3. Have the authors made all data underlying the findings in their manuscript fully available?

Reviewer #1: Yes

Reviewer #2: Yes

4. Is the manuscript presented in an intelligible fashion and written in standard English?

Reviewer #1: Yes

Reviewer #2: Yes

5. Review Comments to the Author

Reviewer #1: Shahapal et al. paper utilizes a FAM19A5 KI mouse model to investigate protein’s function in brain development and behavior. This model exhibits a partial reduction in protein levels measured by immune blotting and in the CSF. However, this reduction does not appear to impact overall brain morphology or development. The authors report alterations in dendritic spine distribution, with a shift toward immature forms. Additionally, they observe lower body weights and other behavioral changes, including hyperactivity and delayed fear response. The authors conclude that while FAM19A5 doesn't influence overall brain structure, it plays a role in neural connectivity by regulating spine maturation. While the explored idea is interesting, I save one major and a few minor concerns that must be addressed.

Major comment:

The authors observed no overall impact on brain morphology, hence no impact on brain development in the FAM19A5 KI mouse model. The only reported impact on brain structure is a shift toward immature spine types, which the authors interpret as a potential disruption in synaptic plasticity, which may underlie low body weight and daily food intake. I believe the authors don’t have enough data to support their hypothesis, given that all other brain parameters were not affected. The authors are required to provide more evidence of disrupted synaptic plasticity in the FAM19A5 KI mouse model.

Minor comments:

1. In line 2 of the results, remove “these” from the phrase “these two amino acids from the translated protein.”

2. Is there a specific reason for the variation in the number of mice used across different experiments? For instance, only 3 mice were used to quantify FAM19A5 levels in the cortex and hippocampus, whereas 6–12 mice were used to quantify protein levels in the CSF (Fig 1).

In Fig 1C, only the levels of non-glycosylated FAM19A5 are affected in the +/LacZ group; however, the authors did not provide any commentary on this observation. Additionally, in Fig 1D, please specify which form of the protein is represented (non-glycosylated, glycosylated, or both). Moreover, in Fig 1E, it is unclear how the protein levels were measured in the CSF. Could you clarify the methodology?

The authors do not provide any data to explain the low levels of FAM19A5 in LacZ KI mice. While they speculate that this may be due to protein degradation, this hypothesis could be tested using a protease inhibitor assay.

MT FAM19A5 exhibited a high binding affinity for LRRC4B in Fig 1E, possibly due to protein overexpression. HEK293 cells seem to have higher MT FAM19A5 levels, especially the non-glycosylated form, compared to protein levels shown in Fig 1C.

In Fig 1E, MT FAM19A5 demonstrated a high binding affinity for LRRC4B, which could potentially be attributed to protein overexpression. Notably, HEK293 cells appear to exhibit higher levels of MT FAM19A5, particularly the non-glycosylated form, compared to the protein levels shown in Fig 1C.

3. In Fig 2, please label "A" as male and "B" as female for clarity. Additionally, it appears that the body weights of males are more affected compared to females; however, the authors have not provided any comments on this observation. Furthermore, is there any available data on female food intake?

Reviewer #2: This manuscript describes the further investigation of FAM19A5-LacZ KI mice which express a carboxy-terminally altered FAM19A5 protein. FAM19A5 is encoded by the TAFA Chemokine Like Family Member 5 (TAFA5) gene. TAFA proteins function as brain-specific chemokines or neurokines that act as regulators of immune and nervous cells. First, the expression level of the altered FAM19A5 and binding to its receptor LRRC4B where analyzed revealing residual expression of about 35% and less affinity, respectively. Phenotypical observation of revealed smaller body size and reduced body weight, altered spine morphology of several cortical neurons, and subtle behavioral changes related to FAM19A5-LacZ KI mice activity and fear conditioning.

The experimental approach is sound and adds to further characterization of FAM19A5-LacZ KI mice. These mice may serve as a model for considerable but not complete loss of FAM19A5, however, the carboxy terminal alteration does not permit the distinction between loss-of-function versus altered, compromised function (receptor binding). In this respect, comparison of FAM19A5-LacZ KI mice with heterozygous FAM19A5 mice may be helpful. The results are clearly described and carefully interpreted. The statistical analysis appears correct.

Minor points:

- the introduction lacks a description of the TAFA5 gene and FAM19A5 with potential structure and functions saving the reader to dig in the original literature.

- the quality of the Western blot shown in Fig 1C is not optimal

- the ELISA shown in Fig 1G is based only on n=2

- increased activity is too strongly interpreted as reduced anxiety

Additional note: supplies Fig 1 is very nice

6. PLOS authors have the option to publish the peer review history of their article (what does this mean? ). If published, this will include your full peer review and any attached files.

**Do you want your identity to be public for this peer review?** For information about this choice, including consent withdrawal, please see our Privacy Policy .

Reviewer #1: No

Reviewer #2: No

---

## [Author Response · Author response to Decision Letter 1]

27 Mar 2025

Response to Reviewers

Editor

In Figure 5, the authors made observations on the difference of different types of dendritic spines between the wildtype and the mutant mice. To help the readers to understand the figure, the authors should provide an image illustrating the different type of spines such as filopodia, mushroom and branched spines with clear labeling. The submitted figures were shown in an order from Fig7 to Fig1, which is not the usual order for manuscript submission. Please re-order the figure sequence from Fig1 to Fig7 during the revision.

Thank you for your suggestion. As mentioned in the Methods section, we referenced established criteria from a previous study to classify dendritic spines. We have incorporated representative images of spines that meet these criteria, with clear labeling, into Figure 5.

Figure 5 Alteration in dendritic spine type in the adult FAM19A5-LacZ KI mouse brain. (A-B) Classified spine types in (A) apical and (B) basal dendrites of layer 2/3 pyramidal neurons in motor cortex. Representative image of thin/filopodia, mushroom, and branched spine was inserted in (A).

Reviewer #1

Major comment:

The authors observed no overall impact on brain morphology, hence no impact on brain development in the FAM19A5 KI mouse model. The only reported impact on brain structure is a shift toward immature spine types, which the authors interpret as a potential disruption in synaptic plasticity, which may underlie low body weight and daily food intake. I believe the authors don’t have enough data to support their hypothesis, given that all other brain parameters were not affected. The authors are required to provide more evidence of disrupted synaptic plasticity in the FAM19A5 KI mouse model.

We agree with the reviewer's comments. We have observed changes in dendritic spine structure, but we lack direct evidence to support our hypothesis that this may affect synaptic plasticity. For the purpose of clarity and to avoid potential reader misinterpretation, we have modified the sentences and removed the term 'synaptic plasticity' from the revised manuscript. Please refer to lines 386-388 and 400-401 for these updates.

Minor comments:

1. In line 2 of the results, remove “these” from the phrase “these two amino acids from the translated protein.”

Thanks for pointing this out. We have removed ‘these’ in the revised manuscript. Please see lines 247.

2. Is there a specific reason for the variation in the number of mice used across different experiments? For instance, only 3 mice were used to quantify FAM19A5 levels in the cortex and hippocampus, whereas 6–12 mice were used to quantify protein levels in the CSF (Fig 1).

In accordance with IACUC policies, we aimed to minimize animal usage to reduce unnecessary sacrifice. Therefore, we predicted the necessary n-number to achieve statistical significance through pilot experiments and sacrificed animals accordingly. In Figure 1B, it was predicted that statistical significance would not be attainable even with repeated experiments. Conversely, in Figure 1E, it was predicted that statistically significant data could be obtained, leading to the sacrifice of a larger number of animals. In Figure 1D, the n-number was increased from 4 to 6 due to the additional Western blot performed for this revision, as in Figure 1C.

In Fig 1C, only the levels of non-glycosylated FAM19A5 are affected in the +/LacZ group; however, the authors did not provide any commentary on this observation.

Thank you for your insightful comment. In response to a comment by reviewer 2, we obtained higher-resolution Western blot images using fresh brain tissue and a refined method, as shown in revised Figure 1C. These images provided clearer band patterns compared to the previous ones. Consistent with our prior findings, we observed a significant decrease in WT FAM19A5 protein expression from WT to heterozygous and homozygous FAM19A5-LacZ brains. Notably, this reduction was independent of glycosylation status in both heterozygous and homozygous FAM19A5-LacZ brains. As the LacZ insertion does not affect glycosylation sites within the FAM19A5 amino acid sequence, both WT and mutant FAM19A5 exhibit comparable glycosylation, eliminating any potential effect attributable to this modification.

In addition, in this gel image, the mutant FAM19A5 from the FAM19A5LacZ/LacZ mouse brain displayed a higher molecular weight compared to wild-type FAM19A5. We have replaced the previous Figure 1C with these new results, marked the additionally identified band in the figure, and added further explanation in the text. Please refer to lines 255-258 and 271-272of the revised manuscript.

Figure 1C. Western blot analysis of FAM19A5 in whole brain lysate from FAM19A5-LacZ KI mice and WT littermates. Open arrowhead for non-glycosylated FAM19A5, solid arrowhead for glycosylated FAM19A5, open arrow for non-glycosylated, mutant FAM19A5 from FAM19A5LacZ/LacZ mice, and solid arrow for non-specific band.

Additionally, in Fig 1D, please specify which form of the protein is represented (non-glycosylated, glycosylated, or both).

The FAM19A5 antibody we used is capable of detecting both non-glycosylated and glycosylated forms of FAM19A5. Therefore, the data presented in Fig. 1D quantifies the expression levels of both forms of FAM19A5. To provide clarity, we have added a description to the figure caption stating that we quantified both non-glycosylated and glycosylated FAM19A5. This update can be found on lines 258-260 of the revised manuscript.

Moreover, in Fig 1E, it is unclear how the protein levels were measured in the CSF. Could you clarify the methodology?

We apologize for any confusion this may have caused. For Figure 1E, we employed ELISA to quantify FAM19A5 protein levels in the CSF, a procedure detailed in the Materials and Methods section of our previous manuscript. To address this, we have included additional descriptive information in the revised manuscript and expanded upon the ELISA procedure in the Materials and Methods. Please find these updates on lines 137-138 and lines 272-274.

The authors do not provide any data to explain the low levels of FAM19A5 in LacZ KI mice. While they speculate that this may be due to protein degradation, this hypothesis could be tested using a protease inhibitor assay.

We appreciate your suggestion. To explore potential mechanism for the reduced FAM19A5 protein levels observed in the FAM19A5-LacZ KI mouse brain, we conducted in vitro experiments using HEK293 cells overexpressing WT or MT FAM19A5 proteins. Cells were treated with cycloheximide to block protein synthesis or MG132 to inhibit the ubiquitin-proteasome pathway. Protein levels were measured using western blot over time after treatment with cycloheximide or MG132.

Following cycloheximide treatment, both protein levels decreased at a similar rate, with approximately 50% degradation observed after 6 hours. However, the protein accumulation rates differed between wild-type (WT) and mutant (MT) FAM19A5 after MG132 treatment. WT FAM19A5 protein levels increased more markedly than MT protein. This observation suggests that WT FAM19A5 is likely degraded via the ubiquitin-proteasome pathway, the primary mechanism of protein degradation, while MT FAM19A5 may be degraded through alternative mechanisms such as autophagy-lysosomal pathway or other unidentified pathways. Alternatively, the reduced accumulation of MT FAM19A5 could be attributed to decreased synthesis due to high misfolding.

However, given the limitations of in vitro models, it is difficult to directly extrapolate the in vitro experimental results to phenomena occurring in the mouse brain. In the mouse brain, FAM19A5 may exhibit different protein stability mechanisms, as it interacts with various proteins, including LRRC4B, resulting in different folding efficiency.

Nevertheless, as the above discussion is based on speculation, the revised manuscript will only describe the observed differences in protein degradation pathways from the in vitro experiments. Please see lines 276-281 of the revised manuscript for these updates.

Supplementary Fig 1. Time-course changes in FAM19A5 protein degradation in HEK293 cells.

(A-B) Time-dependent degradation of FAM19A5. HEK293 cells overexpressing either wild-type (WT) or mutant (MT) FAM19A5 were treated with 200 �g/ml of cycloheximide. Protein levels were then measured over time by Western blot (A) and quantified (B). (C-D) Time-dependent accumulation of FAM19A5 protein was determined in HEK293 cells overexpressing WT or MT FAM19A5, treated with 10 �M of MG132, a proteasome inhibitor to inhibit protein degradation. FAM19A5 protein accumulation was assessed by Western blot (C) and quantified (D). Note that significant cytotoxicity was observed at 48 hours after MG132 treatment.

MT FAM19A5 exhibited a high binding affinity for LRRC4B in Fig 1E, possibly due to protein overexpression. HEK293 cells seem to have higher MT FAM19A5 levels, especially the non-glycosylated form, compared to protein levels shown in Fig 1C.

In Fig 1E, MT FAM19A5 demonstrated a high binding affinity for LRRC4B, which could potentially be attributed to protein overexpression. Notably, HEK293 cells appear to exhibit higher levels of MT FAM19A5, particularly the non-glycosylated form, compared to the protein levels shown in Fig 1C.

Thank you for your comment. We would like to gently correct one point in your question: the measurement of the affinity between FAM19A5 and LRRC4B is shown in Fig. 1 F and G, not Fig. 1E.

As the reviewer pointed out, the relatively high affinity binding of MT FAM19A5 in the Co-IP experiments is likely an artifact of the overexpression system in HEK293 cells. Therefore, the results in Figure 1F were not quantitatively analyzed. However, we quantitatively measured the affinity of MT and WT FAM19A5 to LRRC4B using ELISA (Figure 1G). This experiment utilized purified FAM19A5 proteins, applying equal amounts of WT and MT proteins. In response to reviewer 2's request, we repeated the ELISA experiment with an increased sample size (n=10 per data point). The results revealed a 12.9-fold difference in affinity between WT and MT proteins (EC50: 2564.0 pM for MT vs. 198.9 pM for WT), which contrasts significantly with the Co-IP results.

Consequently, the revised text clarifies that the small affinity difference observed in the Co-IP assay is likely due to protein overexpression in HEK293 cells, whereas the quantitative ELISA analysis demonstrated a 12.9-fold affinity difference between WT and MT proteins. Please see this update on lines 286-290.

3. In Fig 2, please label "A" as male and "B" as female for clarity. Additionally, it appears that the body weights of males are more affected compared to females; however, the authors have not provided any comments on this observation. Furthermore, is there any available data on female food intake?

Thank you for your suggestion. We have added 'male' and 'female' labels to the y-axis legends of Figure 2A and B, respectively. As suggested, we have clarified in the revised manuscript that males exhibit a greater reduction in body weight compared to females. Please see this update on lines 304-305. Due to the more pronounced body weight reduction in males, we examined food intake only in this group. This has been clarified in the revised manuscript on lines 307-308.

Fig 2. Reduced body weight in FAM19A5-LacZ KI mice. (A) Time-dependent body weight changes in male and (B) female FAM19A5-LacZ KI mice compared to FAM19A5+/+ littermates, male: n=16 (WT+/+), 27 (FAM19A5+/LacZ), 15 (FAM19A5LacZ/LacZ), female: n=11 (WT+/+), 18 (FAM19A5+/LacZ), 11 (FAM19A5LacZ/LacZ). Two-way ANOVA followed by Bonferroni’s multiple comparisons test.

Reviewer #2

Minor points:

The introduction lacks a description of the TAFA5 gene and FAM19A5 with potential structure and functions saving the reader to dig in the original literature.

We appreciate the reviewer's recommendation to expand the description of FAM19A5. Accordingly, we have incorporated additional information regarding the therapeutic potential of FAM19A5, its interaction with LRRC4B, and its involvement in synaptic structure modulation and synapse elimination. This addition, as shown in the revised introduction, provides a more comprehensive overview of FAM19A5's role, especially in relation to Alzheimer's disease. Please find this update on lines 41-48.

The quality of the Western blot shown in Fig 1C is not optimal

Thank you for your comment. To ensure better data quality, we used fresh samples and a refined experimental protocol. As a result, we obtained enhanced image results with clearer bands and pattern changes. Notably, in this gel image, the mutant FAM19A5 from the FAM19A5LacZ/LacZ mouse brain displayed a higher molecular weight compared to wild-type FAM19A5. We have replaced the previous Figure 1C with these new results, marked the additionally identified band in the figure, and added further explanation in the text. Please refer to lines 255-258 and 271-272 in the revised manuscript.

Figure 1C. Western blot analysis of FAM19A5 in whole brain lysate from FAM19A5-LacZ KI mice and WT littermates. Open arrowhead for non-glycosylated FAM19A5, solid arrowhead for glycosylated FAM19A5, open arrow for non-glycosylated, mutant FAM19A5 from FAM19A5LacZ/LacZ mice, and solid arrow for non-specific band.

The ELISA shown in Fig 1G is based only on n=2

Thank you for pointing this out. We agreed that increasing the sample size would yield more reliable results. Therefore, we increased the number of n to 10 for each data point. As a result, the difference in binding affinity remained similar to the previous trend, approximately 12.9-fold (EC50 with MT vs. WT FAM19A5: 2564.0, 198.9 pM). We have replaced and added this new result to the figure and revised the corresponding text in the revised manuscript. Please find this update on lines 263 and 289-290.

Figure 1G. The binding affinity of WT or MT FAM19A5 proteins to LRRC4B was measured using ELISA. n=10 each.

Increased activity is too strongly interpreted as reduced anxiety

We agree with the reviewer's comments. We have toned down or removed interpretations related to anxiety. Please refer to lines 408-409, 449-451 and 460-462 in the revised manuscript.

Additional note: supplies Fig 1 is very nice

Thank you for your kind feedback. We appreciate your positive comment regarding Figure 1.

---

## [Decision Letter · Decision Letter 1]

PONE-D-24-51742R1Partial FAM19A5 deficiency in mice leads to disrupted spine maturation, hyperactivity, and an altered fear responsePLOS ONE

Dear Dr. Seong,

Thank you for submitting your manuscript to PLOS ONE. After careful consideration, we feel that it has merit but does not fully meet PLOS ONE’s publication criteria as it currently stands. Therefore, we invite you to submit a revised version of the manuscript that addresses the points raised during the review process.

Figure1SA westernblot showed a clear splice mark between WT48h and MT0h. Apparently, the results were from two gel runs. If so, they should not be put together into one panel. Separate gel runs should be presented by different panels. Figure 1SB, D, the term "the relative protein expression" should be explained in the method section.

We look forward to receiving your revised manuscript.

Kind regards,

Hualin Fu

Academic Editor

PLOS ONE

Journal Requirements:

Additional Editor Comments:

Figure1SA westernblot showed a clear splice mark between WT48h and MT0h. Apparently, the results were from two gel runs. If so, they should not be put together into one panel. Separate gel runs should be presented by different panels. Figure 1SB, D, the term "the relative protein expression" should be explained in the method section.

Reviewers' comments:

Reviewer's Responses to Questions

**Comments to the Author**

1. If the authors have adequately addressed your comments raised in a previous round of review and you feel that this manuscript is now acceptable for publication, you may indicate that here to bypass the “Comments to the Author” section, enter your conflict of interest statement in the “Confidential to Editor” section, and submit your "Accept" recommendation.

Reviewer #1: All comments have been addressed

2. Is the manuscript technically sound, and do the data support the conclusions?

Reviewer #1: Yes

3. Has the statistical analysis been performed appropriately and rigorously? 

Reviewer #1: I Don't Know

4. Have the authors made all data underlying the findings in their manuscript fully available?

Reviewer #1: Yes

5. Is the manuscript presented in an intelligible fashion and written in standard English?

Reviewer #1: Yes

6. Review Comments to the Author

Reviewer #1: Dear Authors,

Thank you for your effort to address the feedback provided by the reviewer.

The authors have addressed all my previous comments. However, I have a follow-up regarding Fig S1, which was generated in response to my previous comment regarding the low levels of MT FAM19A5 in the lacZ KI mice. The authors used MG132 to explore this pathway, and the results are shown in Figs1.

The authors transfected the MT FAM19A5 protein in HEK 293, yet the protein migrates at the same molecular weight as the WT. Similarly, in Fig 1F, both WT and MT appear to run at the same level. However, in the updated version of Fig 1C, the authors stated that the WB analysis reveals a minor size difference in non-glycosylated MT FAM19A5 due to the additional 8 a.a at the C-term. It remains unclear whether the band represented by the open arrow in Fig1C represents non-glycosylated MT FAM19A5 from FAM19A5 LacZ/LacZ mice.

Additionally, in Fig 1s, it is unclear why, at the 0h-timepoint, the CHX-treated group shows high levels of both WT and MT, while in the MG132-treated group, WT protein levels appear low and MT starts at higher levels. The increased accumulation of WT protein over time may be an artifact due to the initially low signal at 0h. Clarification is also needed on how the blots were quantified as the quantification in panel D doesn’t seem to reflect the WB shown in panel C. How many independent experiments were performed to generate these results?

I suggest combining the two experiments into a single protocol in which cells are first treated with CHX to prevent additional protein synthesis and then with MG132 for 48 hours. This would allow for a more accurate assessment of protein half-life rather than relying solely on relative protein expression. If MG132 stabilizes the MT FAM19A5, it would support the author's hypothesis regarding protein degradation pathways. However, However, this is only a recommendation, the authors may choose to either conduct this revised experiment or clarify the current text accordingly.

Lastly, in Fig 1S A, the WT blot is cropped at the 48h time point, please ensure that the full band is visible.

Line 272, remove the letter s from "(Figs 1C)"

Good luck.

7. PLOS authors have the option to publish the peer review history of their article (what does this mean? ). If published, this will include your full peer review and any attached files.

**Do you want your identity to be public for this peer review?** For information about this choice, including consent withdrawal, please see our Privacy Policy .

Reviewer #1: No

---

## [Author Response · Author response to Decision Letter 2]

14 Jun 2025

We have addressed the questions from the editor and reviewer in the "Response to reviewer" file. All modifications made to the manuscript in response to these comments are highlighted in the "Revised Article with Changes Highlighted" file.

---

## [Editor Report · Decision Letter 2]

Partial FAM19A5 deficiency in mice leads to disrupted spine maturation, hyperactivity, and an altered fear response

PONE-D-24-51742R2

Dear Dr. Seong,

We’re pleased to inform you that your manuscript has been judged scientifically suitable for publication and will be formally accepted for publication once it meets all outstanding technical requirements.

Kind regards,

Hualin Fu

Academic Editor

PLOS ONE

Additional Editor Comments (optional):

The revised manuscript is now suitable for publication.
---

## [Editor Report · Acceptance letter]

PONE-D-24-51742R2

PLOS ONE

Dear Dr. Seong,

I'm pleased to inform you that your manuscript has been deemed suitable for publication in PLOS ONE. Congratulations! Your manuscript is now being handed over to our production team.

Kind regards,

on behalf of

Dr. Hualin Fu

Academic Editor

PLOS ONE